# Alteration of ribosome function upon 5-fluorouracil treatment favors cancer cell drug-tolerance

Gabriel Therizols[1,2,3], Zeina Bash-Imam[1,2,3], Baptiste Panthu[4,16], Christelle Machon[1,2,3,5,6], Anne Vincent[1,2,3], Julie Ripoll[7], Sophie Nait-Slimane[1,2,3], Mounira Chalabi-Dchar[1,2,3], Angéline Gaucherot[1,2,3], Maxime Garcia[1,2,3], Florian Laforêts[1,2,3], Virginie Marcel[1,2,3], Jihane Boubaker-Vitre[8], Marie-Ambre Monet[1,2,3], Céline Bouclier[8], Christophe Vanbelle[1,2,3], Guillaume Souahlia[1,2,3], Elise Berthel[1,2,3], Marie Alexandra Albaret[1,2,3,17], Hichem C. Mertani[1,2,3], Michel Prudhomme[9], Martin Bertrand[9], Alexandre David[8,10], Jean-Christophe Saurin[1,2,3,11], Philippe Bouvet[1,2,3,12], Eric Rivals[7,13], Théophile Ohlmann[4], Jérôme Guitton[1,2,3,6,14], Nicole Dalla Venezia[1,2,3], Julie Pannequin[8], Frédéric Catez[1,2,3,15,18✉] & Jean-Jacques Diaz[1,2,3,15,18✉]

Mechanisms of drug-tolerance remain poorly understood and have been linked to genomic but also to non-genomic processes. 5-fluorouracil (5-FU), the most widely used chemotherapy in oncology is associated with resistance. While prescribed as an inhibitor of DNA replication, 5-FU alters all RNA pathways. Here, we show that 5-FU treatment leads to the production of fluorinated ribosomes exhibiting altered translational activities. 5-FU is incorporated into ribosomal RNAs of mature ribosomes in cancer cell lines, colorectal xenografts, and human tumors. Fluorinated ribosomes appear to be functional, yet, they display a selective translational activity towards mRNAs depending on the nature of their 5'-untranslated region. As a result, we find that sustained translation of IGF-1R mRNA, which encodes one of the most potent cell survival effectors, promotes the survival of 5-FU-treated colorectal cancer cells. Altogether, our results demonstrate that "man-made" fluorinated ribosomes favor the drug-tolerant cellular phenotype by promoting translation of survival genes.

[1] Inserm U1052, CNRS UMR5286 Centre de Recherche en Cancérologie de Lyon, F-69000 Lyon, France. [2] Centre Léon Bérard, F-69008 Lyon, France. [3] Université de Lyon 1, F-69000 Lyon, France. [4] CIRI-Inserm U1111, Ecole Normale Supérieure de Lyon, Lyon F-693643, France. [5] Laboratoire de chimie analytique, Faculté de pharmacie de Lyon, 8 avenue Rockefeller, F-69373 Lyon, France. [6] Laboratoire de biochimie et de pharmaco-toxicologie, Centre hospitalier Lyon-Sud – HCL, F-69495 Pierre Bénite, France. [7] LIRMM, UMR 5506, University of Montpellier, CNRS, Montpellier, France. [8] IGF, Univ. Montpellier, CNRS, INSERM, Montpellier, France. [9] Department of Digestive Surgery, CHU Nimes, Univ Montpellier, Nimes, France. [10] IRMB-PPC, Univ Montpellier, INSERM, CHU Montpellier, CNRS, Montpellier, France. [11] Department of Endoscopy and Gastroenterology, Pavillon L, Edouard Herriot Hospital, Lyon, France. [12] Ecole Normale Supérieure de Lyon, Lyon, France. [13] Institut Français de Bioinformatique, CNRS UMS 3601 Évry, France. [14] Laboratoire de toxicologie, Faculté de pharmacie de Lyon, Université de Lyon, 8 avenue Rockefeller, F-69373 Lyon, France. [15] Institut Convergence PLAsCAN, F-69373 Lyon, France. [16] Present address: Inserm U1060, CARMEN, F-69310 Pierre Bénite, France. [17] Present address: Department of Translational Research and Innovation, Centre Léon Bérard, 69373 Lyon, France. [18] These authors jointly supervised this work: Frédéric Catez, Jean-Jacques Diaz. ✉email: frederic.catez@lyon.unicancer.fr; jean-jacques.diaz@lyon.unicancer.fr

Partial response to chemotherapy leads to disease resurgence. Upon treatment, a subpopulation of cancer cells, called drug-tolerant persistent cells, displays a transitory drug tolerance that leads to treatment resistance[1,2]. Though drug-tolerance mechanisms remain poorly understood, they have been linked to non-genomic processes, including epigenetics, stemness and dormancy[2–4].

Translation regulation plays a major role in controlling gene expression and contributes to disease emergence including cancer[5,6]. Within ribosomes, ribosomal RNAs (rRNAs) play a central role in the translation process, by monitoring codon:anti-codon recognition, coordinating ribosomal subunit activity and catalyzing peptide-bond formation through their ribozyme activity. rRNAs contain over 200 naturally occurring chemical modifications, which stabilize rRNA structures and create additional molecular interactions not provided by non-modified nucleotides[7–9]. Chemical modifications of rRNAs were shown to directly contribute to translational regulation[6,10,11]. We, and others, showed that rRNA chemical modifications contribute to the fine-tuning of ribosome functions and to the modulation of ribosome translational activity of ribosomes in cancer cells[12–15]. 5-fluorouracil (5-FU) is the most widely used chemotherapy in cancer treatment. 5-FU efficacy is partial and often associated with resistance[16]. While discovered and used as an inhibitor of DNA replication, 5-FU alters all RNA pathways[16–20]. Indeed, 5-FU treatment results in 5-fluorouridine (5-Urd) incorporation into various types of cellular RNAs including the precursor of rRNA[16]. However, the consequences of 5-FUrd incorporation into ribosomal RNA precursor on ribosome production and functioning have so far not been analyzed, neither has its impact on cellular phenotype. Here, we show that 5-FU is incorporated into rRNAs of mature ribosomes in several models including cancer cell lines, colorectal mouse xenografts, and human colorectal tumor samples. 5-FU containing ribosomes appear to be functional, yet, they display a selective translational activity towards mRNA subsets depending on the nature of their 5'-untranslated region. We find that upon 5-FU treatment, translation of the mRNA of the pro-survival IGF-1R gene is sustained, and promotes the survival of 5-FU-treated colorectal cancer cells. Altogether, our results demonstrate that fluorinated ribosomes favor the drug-tolerant cellular phenotype by promoting the translation of survival genes.

## Results

**Ribosome production is partially maintained upon 5-FU treatment.** Previous work indicated that at a high concentration, 5-FU alters ribosome biogenesis without inhibiting pre-rRNA synthesis[20,21]. To further investigate this, we treated colorectal cancer HCT116 cells with clinically relevant concentrations of 5-FU (10–50 μM)[22,23], which resulted in growth inhibition and cell death ([24] and Supplementary Fig. 1a). Within this concentration range, 5-FU treatment resulted in enlarged nucleoli, absence of nucleolar cap formation and absence of dispersion of nucleolar markers, as opposed to cells treated with the RNA Pol I inhibitor actinomycin D (Fig. 1a and Supplementary Fig. 1b, c). Such nucleolar restructuring reveals an alteration of ribosome biogenesis albeit without pre-rRNA synthesis inhibition, and was confirmed by transmission electron microscopy (TEM) (Supplementary Fig. 1d). Consistently, 47 S/45 S pre-rRNA levels, analyzed by Northern blotting and RNA fluorescent in situ hybridization (FISH), were unchanged following 5-FU treatment confirming that 5-FU did not affect RNA Pol I activity (Fig. 1b and Supplementary Figs. 1e, 2a, b).

Northern blot analysis also confirmed that ribosome maturation at post-transcriptional steps was altered, and revealed that the pre-rRNA processing was impaired at the cleavage stage at site 2 (Supplementary Fig. 2a–c). Yet, despite this effect, the late pre-rRNA intermediates leading to 18 S and 28 S rRNA were still detected (Supplementary Fig. 2c) suggesting that ribosome production was in part maintained. This was confirmed by [32P] pulse-chase experiments that showed that ribosomes are produced at significant levels for up to 48 h under 5-FU treatment (Fig. 1c and Supplementary Fig. 2d). Thus, at a clinically relevant concentration of 5-FU, each step of ribosome processing is able to proceed, despite the stringent quality control, thus allowing ribosome production to be maintained at a substantial level.

**5-FU incorporates into ribosomes.** 5-FU was previously shown to be converted to 5-FUTP and incorporated into RNAs[16]. We therefore wondered whether ribosomes produced and exported to the cytoplasm in treated cells contained 5-FUrd within their rRNAs. To test this, we developed a quantitative liquid chromatography—mass spectrometry—high-resolution mass spectrometry (LC-MS-HRMS) approach that allowed us to determine the number of 5-FUrd incorporated into rRNA of cytoplasmic ribosomes purified at high stringency on a 500 mM KCl sucrose cushion (Fig. 2a, see methods for details[25]). We found that HCT116 ribosomes contained significant amounts of 5-FUrd, ranging from 7 to 14 5-FUrd molecules per ribosome following 24 h of treatment with 5–100 μM of 5-FU (Fig. 2b). We performed two experiments to rule out that the 5-FUrd signal came from nonribosomal RNA. First, we measured 5-FUrd from gel-purified 18 S and 28 S rRNA (Supplementary Fig. 3a) to isolate 18 S and 28 S rRNA from the other cellular RNAs by cutting out the corresponding bands from the gel. Second, because direct purification of ribosomes on a 500 mM KCl sucrose cushion might result in minor mRNA contamination, we measured 5-FUrd from ribosomal subunits dissociated with puromycin (Supplementary Fig. 3b, c). Our data show that puromycin treatment of the ribosomal subunit fully dissociated a small number of subunits that were still assembled after direct purification on the 500 mM KCl sucrose cushion (Supplementary Fig. 3b). Yet the number of 5-FUrd per ribosome was identical in ribosomal subunits separated with puromycin compared to ribosomal subunits directly purified on the 500 mM KCl sucrose cushion (Supplementary Fig. 3c), indicating that the data obtained by our method reflects 5-FUrd incorporation into rRNA. Next, we extended the analysis to additional cancer cell lines: (i) CRC cells characterized by different molecular profiles reflective of the pathology displaying the different combination of KRAS, BRAF and TP53 mutations, as well as the microsatellite instability (MSI) and CIN statuses (Supplementary Table 1) and (ii) cell lines from triple-negative breast cancer and (ii) pancreatic cancer. 5-FUrd was incorporated into rRNA of cytoplasmic ribosomes purified from all tested cell lines after 24 h of treatment with 10 μM 5-FU (Fig. 2c). Altogether, these data demonstrate that upon 5-FU treatment, ribosomes containing fluorinated rRNA are fully assembled and exported to the cytoplasm, revealing that presence of 5-FUrd is tolerated by the quality control systems of the cell.

Next, we investigated whether fluorinated ribosomes could be found within tumors in vivo. First, we analyzed rRNA from HCT116 xenografts established in nude mice. 5-FU treatment efficacy was evidenced by a decrease in tumor growth (Supplementary Fig. 3d). 5-FUrd was detected in mature rRNA purified from tumor cells collected after the last treatment at levels close to those observed in cultured cells (Fig. 2d). We extended this observation to xenografts similarly established with HT29 and SW480 cell lines (Fig. 2e, f). Thus, 5-FU incorporation into ribosomes can be replicated in a common

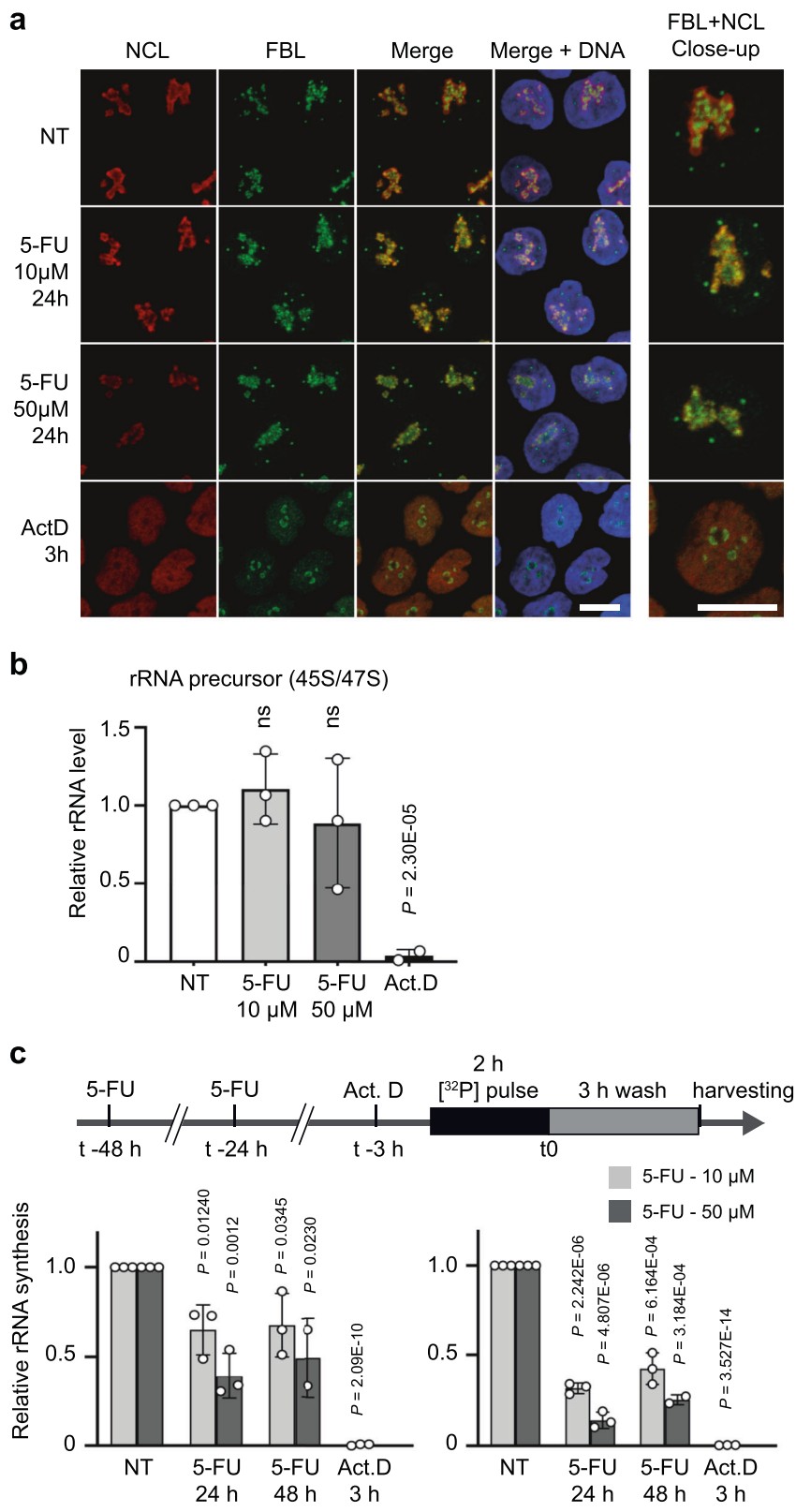

xenografted animal model. Finally, we analyzed rRNA of colorectal tumor cells from patients treated with 5-FU-based therapies, using large RNA quantities to optimize detection (Supplementary Fig. 3e). Of the 5 samples tested from 5-FU-treated patients, 5-FUrd was detected in rRNA from 2 patients (3.80 and 4.50 5-FUrd per ribosome respectively; Fig. 2g), a patient receiving no 5-FU served as a negative control. Altogether, these data show that 5-FUrd is largely incorporated into rRNA of cells treated with 5-FU, and that 5-FU-based chemotherapy leads to the production of fluorinated ribosomes within tumor cells both in animal models and in humans.

**Fig. 1 Ribosome production is maintained in 5-FU-treated cells.** HCT116 cells were treated with 5-FU at 10 μM or 50 μM for 24 h or 48 h or with actinomycin D (Act.D) for 3 h as a reference of rRNA synthesis inhibition. **a** Morphology of nucleoli analyzed by immunofluorescent detection of nucleolar markers nucleolin (NCL, red) and fibrillarin (FBL, green). Nuclei were stained with Hoechst (blue). Scale bar = 10 μM. Images are representative of three independent experiments. **b** Pre-rRNA synthesis analyzed by detection of 47 S/45 S rRNA precursor levels by Northern blotting. Data are expressed as mean values +/- s.d. of independent experiments ($n = 3$). **c** Rate of 28 S and 18 S rRNAs production analyzed by isotope pulse labelling. Radioactivity was measured for each rRNA and normalized against ethidium bromide. Data are expressed as mean +/- s.d. of independent experiments ($n = 3$). Results of unpaired two-tailed $t$ test are indicated as nonsignificant (ns) or with the $P$ value. Source data are provided as a Source Data file.

**Altered translation by fluorinated ribosomes**. Because rRNAs and their post-transcriptional chemical modifications play a central role in ribosome functioning, and because 5-FU induces changes in translational regulation[24,26,27], we postulated that fluorinated ribosomes may display modified translational activity. To investigate this, we first considered whether fluorinated ribosomes could be recruited onto mRNA during translation, by analyzing the rRNA 5-FUrd content in actively translating ribosomes isolated by sucrose gradient (Fig. 3a and Supplementary Fig. 4). 5-FU was readily detected in actively translating ribosomes (i.e. polysomal ribosomes), demonstrating that fluorinated ribosomes can engage in translation (Fig. 3a). Next, we evaluated whether the incorporation of 5-FU into rRNA impacts the translational capacity of ribosomes. We used our recently developed in vitro hybrid translation assay[14,28], in which only ribosomes have been exposed to 5-FU, in order to evaluate the activity of purified fluorinated ribosomes in a controlled setting (Fig. 3b). To gain insight into the changes in ribosome activity the translational capacity of fluorinated ribosomes was assessed using a set of luciferase reporter mRNAs, representing different functional mRNAs and whose translation relies on different 5′UTR: (i) mRNAs of two housekeeping genes containing short 5′UTR from globin and GAPDH mRNAs, (ii) mRNA of two cancer-promoting genes containing long and structured 5′UTR from IGF-1R and c-Myc mRNAs, and (iii) mRNA of two viral genes containing long and structured uncapped 5′UTR from cricket paralysis virus (CrPV) and encephalomyocarditis virus (EMCV), which initiate translation through an internal ribosome entry site (IRES). The results showed first that fluorinated ribosomes were not impaired for translation. Second that they displayed a selective translation initiation efficacy that differed from that of control ribosomes, and varied according to the nature of the 5′UTR upstream of the reporter mRNA used (Fig. 3c). Indeed, globin and GAPDH were less efficiently translated, a result that is consistent with lower overall protein synthesis in 5-FU treated cells ([24], and Supplementary Fig. 5a). Moreover, reporter mRNAs containing *IGF-1R* and *c-Myc* 5′-UTR were more efficiently translated by fluorinated ribosomes. These differences suggest that translation efficiency varies according to the nature of the 5′UTR, indicating that the initiation step of translation differs for fluorinated ribosomes compared to normal ribosomes. To consolidate this hypothesis, translation was tested on a mRNA carrying the CrPV intergenic IRES, an element that directly binds to the ribosome and initiates translation without any cellular translation initiation factors (eIFs). Fluorinated ribosomes displayed a decrease in translational activity on CrPV mRNA, strongly supporting that fluorinated ribosomes are structurally or functionally different (Fig. 3d). This defect in translation initiation from the CrPV intergenic IRES was not strictly related to cap-independent initiation mechanisms since fluorinated ribosomes were more efficient at translating an EMCV IRES containing mRNA, another cap-independent translation initiation model (Fig. 3d).

Next, we focused on *IGF-1R* 5′UTR which is one of the longest 5′UTR in the human genome and contains several regulatory elements, the activity of which could be influenced by 5-FU,

including an IRES element[29,30]. First, we mapped the IRES element of *IGF-1R* 5′UTR by conducting a series of deletions (Supplementary Fig. 5b, c) and localized it in the last 84 nucleotides (region 954-1040), and then verified that no IRES activity was carried out by the 1-953 region, an observation consistent with the previous studies[29]. When evaluated in 5-FU treated cells, the IRES activity of *IGF-1R* 5′UTR was increased both when the full-length 5′UTR or only the minimal IRES (region 954-1040) were used (Fig. 3e). In contrast no effect was observed for the 1-953 region (ΔIRES). In addition, *c-MYC* 5′UTR IRES activity was not impacted (Supplementary Fig. 5d), suggesting that the activity of some IRES elements is not impacted by 5-FU and that regulation of *c-MYC* translation involves another mechanism. To determine whether the IRES element could play a dominant role in *IGF-1R* translation regulation, we tested the same construct using a monocistronic assay, where the 5′UTR is directly placed at the 5′ of the reporter mRNA and capped (Fig. 3f). We found that both the full-length 5′UTR and the minimal IRES had significant increased translational activity upon 5-FU treatment, and that the activity of the 1-953 region of *IGF-1R* 5′UTR and that of *GAPDH* 5′UTR were not significantly affected by 5-FU, although we observed a tendency towards a decrease at high 5-FU concentrations (Fig. 3f).

Altogether, these experiments demonstrate that 5-FU incorporation into rRNA modifies the ability of ribosomes to initiate mRNA translation from different 5′UTR. Taken together, they highlight that fluorinated ribosomes might contribute to 5-FU-induced translational reprograming that we previously observed[24].

**IGF-1R promotes 5-FU drug tolerance**. The data above suggest that fluorinated ribosomes favor translation of selected mRNAs, including genes such as *IGF-1R* and *c-Myc*, that may promote early cell survival and lead to resistance[4,31,32]. We focused on *IGF-1R*, a gene that plays a major role in tumorigenesis and whose contribution to cell survival has been largely demonstrated in various models including colorectal cancer[31,33–35]. Because 5-FU treatment induces a decrease in global protein synthesis ([24], Supplementary Fig. 5a), we initially evaluated whether *IGF-1R* mRNA translation was also impacted by 5-FU treatment in HCT116 cells. *IGF-1R* mRNA translation efficacy was assessed by measuring the recruitment of cytoplasmic mRNA into the heavy polysome fraction of control and 5-FU-treated cells (Fig. 4a and Supplementary Fig. 6). Our data show that the fraction of *IGF-1R* mRNAs associated with heavy polysomes was maintained in 5-FU treated cells, while that of actin and GAPDH mRNAs decreased, indicating that translation of *IGF-1R* mRNA is selectively favored compared to housekeeping mRNAs, a result consistent with in vitro translation data obtained with reporter mRNAs (Fig. 3c). At the protein level, IGF-1R increased compared to relatively stable levels of Actin, GAPDH and Histone H3 proteins after 24 h and 48 h of 5-FU treatment (Fig. 4b, c). Next, to determine whether the IGF-1/IGF-1R pathway contributes to the survival of CRC cells exposed to 5-FU, cells were first treated with 5-FU for 24 h or 48 h, and were subsequently treated with

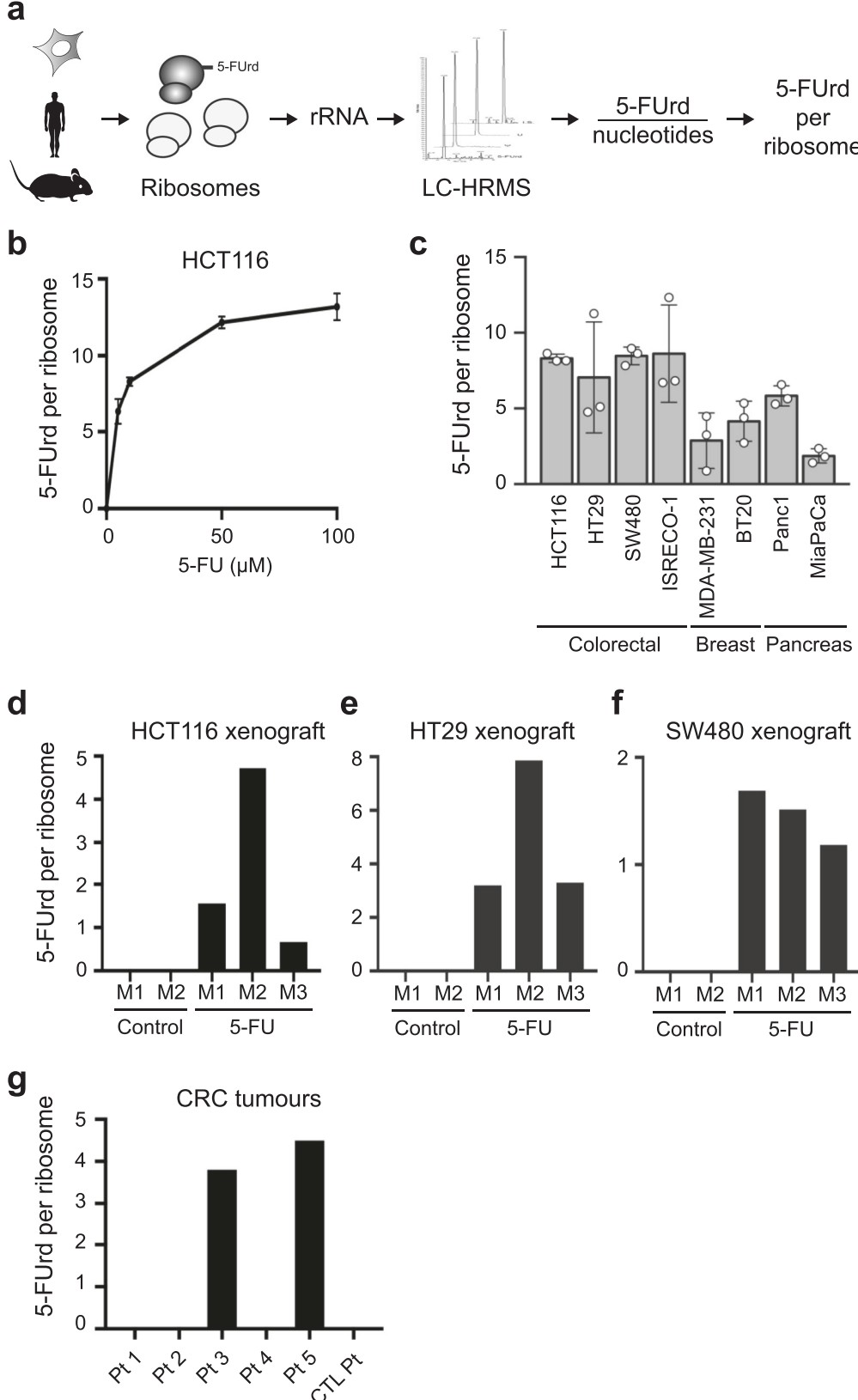

IGF-1. Cell proliferation was monitored over 5 days, and revealed that, while IGF-1 had no impact on control cells, it improved the growth of cells treated with 5-FU (Fig. 4d, e, and Supplementary Fig. 7a). To further validate our findings, HCT116 cells were co-treated with 5-FU and the IGF-1R kinase inhibitor NVP-AEW541[36], and cell response was monitored by viability assay

(Fig. 4f). Inhibition of IGF-1R induced a stronger decrease in cell survival when cells were co-treated with 5-FU compared to untreated cells demonstrating that 5-FU treated cells relied more on an active IGF-1/IGF-1R pathway than untreated cells, and that the IGF-1/IGF-1R pathway is necessary for optimal cell tolerance to 5-FU. Overall, our results unveil that the IGF-1/IGF-1R

**Fig. 2 5-FU is incorporated in ribosomes of cell lines and tumours. a** Schematic representation of 5-FUrd incorporation into ribosomes determined using liquid chromatography—mass spectrometry—high-resolution mass spectrometry (LC-HRMS). **b,** HCT116 cells were treated for 24 h with 5 to 100 μM 5-FU and 5-FUrd incorporation was determined as in a. Data are expressed as mean +/- s.d. of independent experiments (n = 3). **c,** Indicated cell lines were treated for 24 h with 10 μM of 5-FU and incorporation of 5-FUrd into rRNA was determined as in a. Data are expressed as mean +/- s.d. of independent experiments (n = 3). **d–f, d,** HCT116, **e,** HT29 or **f,** SW480 cells were xenografted into nude mice, and mice were treated with 50 mg/kg of 5-FU twice a week (5-FU) or with PBS (Control). Incorporation of 5-FUrd into rRNA was determined as in a, from gel-purified rRNA. Data are values for individual animals (noted M1 to M3) (n = 1). **g,** rRNA were purified from total RNA extracted from colorectal cancer samples. Incorporation of 5-FUrd into rRNA was determined as in a. Pt = sample from 5-FU treated patient, CT Pt = sample from patient not treated with 5-FU. n = 1 for each sample. Source data are provided as a Source Data file.

pathway plays a role in the survival of a cell subpopulation upon 5-FU treatment, and strongly support that the 5-FU driven maintenance of IGF-1R synthesis contributes to this mechanism.

**5-FU treatment leads to translational upregulation of cell survival-associated mRNAs.** Next, we wondered whether the changes in translation induced by 5-FU could promote cellular functions associated with cell survival or drug tolerance. The impact of 5-FU on cellular mRNA translation was evaluated by polysome profiling on cells treated with 10 μM or 50 μM 5-FU for 24 h (Supplementary Fig. 8a, Supplementary Fig. 9). Briefly, mRNAs associated with actively translating ribosomes (i.e. polysomes) were purified (Supplementary Fig. 8b) and analyzed by RNA-Seq to compare the polysomal level of each transcript between 5-FU treated and untreated cells (Fig. 5a, b, Supplementary Data 1). We found that 5-FU treatment altered translation for a fraction of mRNA, representing about 7 % and 10 % of analyzed mRNAs, in 10 μM and 50 μM 5-FU treated cells respectively (Fig. 5a, b), with a majority of translationally altered mRNA being upregulated (10 μM, 702 up vs. 275 down; 50 μM: 937 up vs. 477 down, corresponding to 72% and 66% respectively). An increased number of differentially translated mRNAs was observed between 10 μM and 50 μM 5-FU conditions (977 vs 1414), most of these mRNAs being common to these two conditions, indicating a dose-dependent response (Fig. 5c). A Gene Ontology (GO) enrichment analysis performed on all translationally altered mRNAs (Fig. 5d, Supplementary Fig. 8c and Supplementary Data 2) revealed similar GO-term enrichment for 10 μM and 50 μM treated cells. As expected, enrichment in p53 pathway, DNA damage response and apoptosis were significantly represented, which reflects the well-known cytotoxic activity of 5-FU. Next, a clustering of GO terms, which groups GO terms related to the same functions (Fig. 5e, f), revealed that in response to 50 μM 5-FU, the translationally downregulated mRNAs mainly correspond to gene classes reflecting cell proliferation, such as cell cycle or cell and organelle biogenesis; an observation also consistent with a stress-induced proliferation arrest (Fig. 5e, Supplementary Data 3). Interestingly, a majority of translationally altered mRNAs were upregulated both in 10 μM and 50 μM 5-FU conditions (Fig. 5a, b). These translationally upregulated mRNAs are associated with biological functions promoting cell-survival such as metabolism, cell communication, signal transduction and unexpectedly cell phenotype changes (e.g., development, morphogenesis or cell differentiation) (Fig. 5f and Supplementary Data 3). This observation highlights that, at the translational level, while a stress response associated with cell cycle arrest and cell death is observed, pro-survival-associated pathways are concomitantly upregulated. Therefore, we specifically searched for genes that may contribute to cell survival. To this end, we set-up a list of genes promoting cell survival and tumor progression including genes associated with either mutations or copy number alterations in CRC, or associated with negative regulators of apoptosis and positive regulators of cell cycle functions. Interestingly, more mRNAs encoding survival proteins were found in

the translationally upregulated mRNA category than in the translationally downregulated one, both in 10 μM and 50 μM 5-FU conditions (10 μM, 27 up vs. 18 down; 50 μM: 38 vs 30, Supplementary Data 4). Indeed, the cellular functions of the translationally upregulated mRNAs survival genes were mainly negative regulators of apoptosis, positive regulators of cell cycle and cell signaling factors (Supplementary Data 5). Altogether, these data indicate that 5-FU treatment results in an unexpected increase in translation of mRNAs encoding proteins with cell survival functions, which could contribute to the observed drug tolerance of a subset of cells.

## Discussion

In this study, we reveal that the pyrimidine analogue 5-FU is incorporated into ribosomes in vitro and in vivo, including in human tumors. We used a LC-HRMS method that we recently developed[25] in order to quantitate the level of incorporation of 5-FUrd into a defined RNA molecule. This approach allowed us to demonstrate that 5-FUrd is incorporated into ribosome at significant levels, showing that cells can tolerate the production of non-natural ribosomes. This finding was unexpected because ribosome assembly and maturation require multiple rRNA-rRNA and rRNA-protein interactions and rRNA folding that could be limited or inhibited by the presence of 5-FUrd, and because these steps are under stringent quality-control that induces the degradation of unproperly folded and assembled rRNAs[37], as evidenced by the decrease in the level of the late pre-rRNA species that we report in this study. It can be anticipated that 5-FUrd would be enriched only in rRNA regions of the mature ribosomes where its presence does not significantly inhibit rRNA maturation and folding. As a result, cytoplasmic functional ribosomes contained up to 15 molecules of 5-FUrd per ribosome, a number likely underestimated since only a fraction of the ribosome population was renewed within the time frame of our experiment. We were able to observe 5-FUrd in rRNA of 2 of 5 human CRC samples, highlighting that 5-FU incorporation into tumor ribosomes can occur in patients. We ruled out that the lack of detection of 5-FUrd in 3 of the samples was due the threshold of sensitivity of our assay. It is possible that incorporation takes place only in some tumors, depending on tumor cell metabolism or other molecular traits that remain to be determined. In addition, other parameters might influence 5-FU incorporation and detection such as the chemotherapy regimen or the delay between treatment and surgery. For these reasons, and because our analysis was performed on a small number of samples, we cannot at this stage extrapolate the frequency of incorporation. Most importantly, it remains to establish whether there is a link between 5-FU incorporation into tumor cell ribosomes, tumor response to therapy and disease outcome. While the addition of fluorine into rRNA results in a non-natural modification, and could be anticipated as deleterious, we found that fluorinated ribosomes are functional as they engage in translation. However, their activity is altered and displays a selective ability to initiate mRNA translation according to the nature of its 5′UTR. The

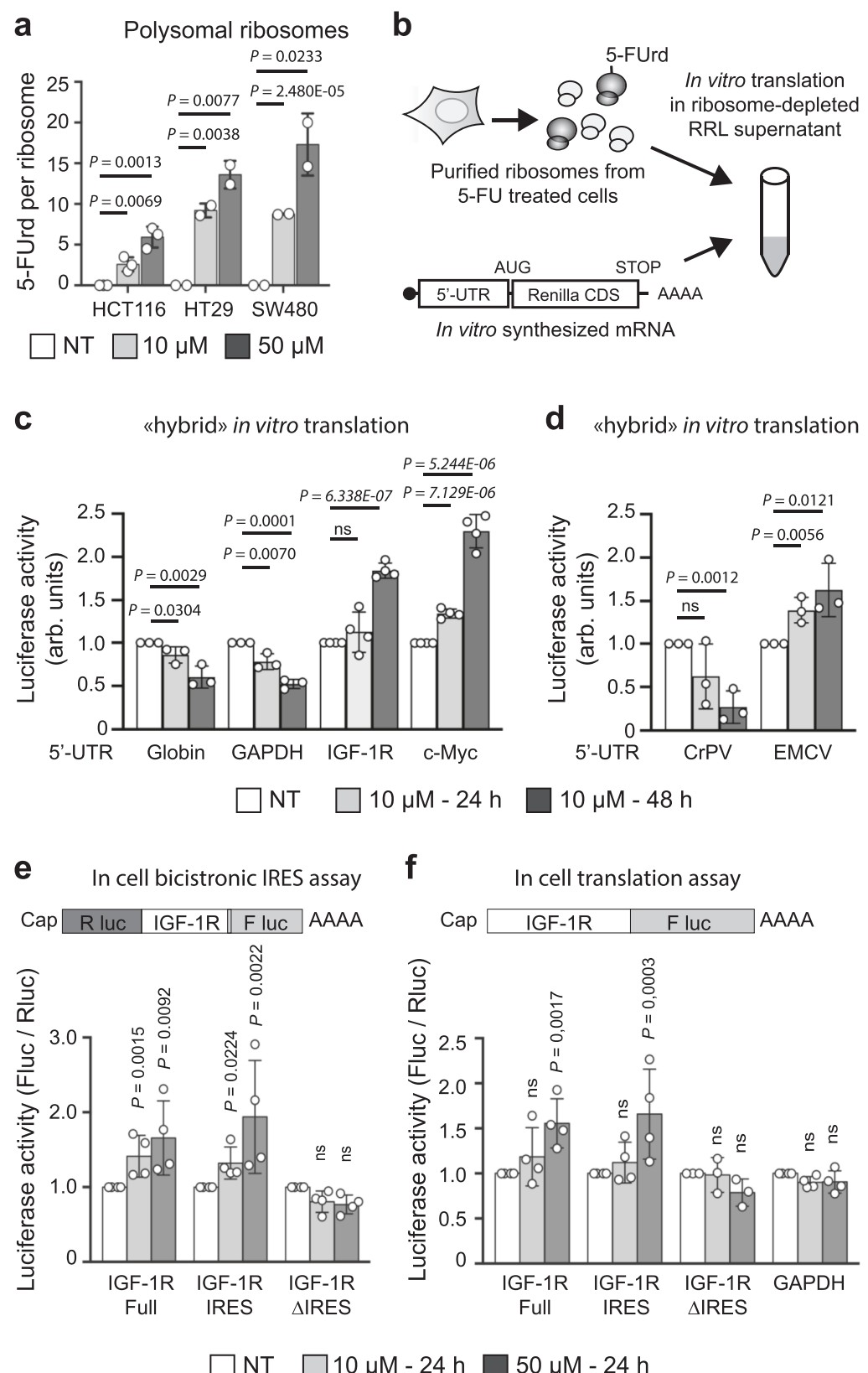

finding that fluorine appears to modify the functioning of rRNA is not unexpected since chemical modifications of rRNA including 2′-O-methylation and pseudouridylation were shown (i) to contribute to translational regulation and efficiency[13–15,38], and (ii) to establish molecular interactions that cannot be provided by non-modified ones[7,9]. The effect we observe on

translation initiation suggests that 5-FUrd incorporation is enriched at or close to regions of the ribosome that are critical for the interaction of the ribosome with mRNAs or translation factors, or in the functional domains of the ribosome, such as the A, P and E sites. Thus, if present in particular regions of the ribosome, even a limited number of 5-FUrd molecule could greatly influence the

**Fig. 3 Fluorinated ribosomes display altered translational properties. a** HCT116, HT29 and SW480 cells were treated for 24 h with either 10 μM or 50 μM 5-FU, and translationally active ribosomes were purified from the polysomal fraction. Incorporation of 5-FUrd was measured by LC-HRMS. Data are expressed as mean +/- s.d of independent experiments ($n = 3$). **b** Schematic representation of the hybrid in vitro translation assay used in **c** and **d**. **c** and **d** Ribosomes were purified from HCT116 cells treated with 10 μM 5-FU for 24 h or 48 h, and their translational activity was evaluated using the hybrid in vitro translation assay. Translation efficacy was evaluated on luciferase reporter mRNA containing the 5′-untranslated region (5′-UTR) of the indicated gene. CDS = coding sequence. Values are units of Renilla luciferase activity normalized against the untreated (NT) condition. **c**, Evaluation on capped mRNA containing the 5′UTR of actin, GAPDH, IGF-1R and c-Myc genes. **d** Evaluation on uncapped mRNA containing the IRES element from the cricket paralysis virus (CrPV) and the encephalomyocarditis virus (EMCV). Data are expressed as mean +/- s.d. of independent experiments ($n = 3$). **e** IRES activity of IGF-1R 5′UTR and deletions was evaluated by transfection of a bicistronic reporter vector represented on top of the figure. IRES activity is monitored as the ratio of firefly luciferase (Fluc) activity over Renilla luciferase (Rluc) activity. HCT116 cells were transfected for 24 h and then treated for 24 h with either 10 μM or 50 μM 5-FU. Data are expressed as mean +/- s.d. of independent experiments ($n = 4$). **f** Translation initiation activity of IGF-1R 5′UTR and deletions and of GAPDH 5′UTR was evaluated by co-transfection of a reporter vector represented on top of the figure with a renilla reporter vector for normalization. Translation activity is monitored as the ratio of firefly luciferase (Fluc) activity over Renilla luciferase (Rluc) activity. HCT116 cells were transfected for 24 h and then treated for 24 h with either 10 μM or 50 μM 5-FU. Data are expressed as mean +/- s.d. of independent experiments ($n = 4$). Results of unpaired two-tailed $t$ test are indicated as nonsignificant (ns) $p < 0.05$ (*), $p < 0.01$ (**), $p < 0.001$ (***), and $p < 0.0001$ (****). Source data are provided as a Source Data file.

ribosome behavior. For instance, in the case of IRES-dependent translation, 5-FUrd may be selectively enriched in rRNA regions close to RPS25, RPL40 or RACK1 proteins that were shown to participate in IRES-dependent translation[39–41], or within 18 S rRNA regions proposed to support IRES translation, such as the expansion segment ES7[42] or the 959-964 region, which was proposed to interact with IGF-1R mRNA[43]. New tools will be required to finely map the location of 5-FUrd within rRNA and improve our understanding of its impact on ribosome functioning at the atomic level.

We previously described a major translational reprograming induced by 5-FU in colorectal cancer cells, that we have linked to a miRNA-based mechanism[24]. Here we describe the 5-FU induced translational reprograming by RNA-Seq-based polysome profiling. Upon 5-FU exposure, the majority of translationally altered mRNAs are upregulated, indicating that 5-FU allows maintenance or increased translation of these mRNAs. The fluorination of rRNA that we describe herein represents an additional mechanism by which 5-FU contributes to translational reprograming of treated cells[24]. In addition to miRNA-based translation regulation, IRES-dependent translation appears as another mechanism, which 5-FU can modulate. IRES-dedicated genome-wide assay will be necessary to evaluate which IRES elements might be sensitive to 5-FU[44]. It is likely that other mechanisms are involved, such as 5-FUrd incorporation into mRNAs and tRNAs.

We determined that the 5-FU altered translational machinery contributes to maintaining the expression level of the *IGF-1R* gene, thus promoting cell survival. This suggests that the cytotoxic efficiency of 5-FU may be improved if fluorinated ribosome production is prevented, an approach that could be effectively tested using the recently developed ribosome biogenesis inhibitors, for which anti-cancer activities are being unveiled[45–47].

Drug tolerance is a critical phase as it represents a window of opportunity for genomic and non-genomic events to take place and provides cells with a drug-resistant phenotype. We show that sustained IGF-1R synthesis is a significant factor for cell survival upon 5-FU treatment. Surprisingly, our data indicate that 5-FU sensitized cells to IGF-1/IGF-1R pathway, as 5-FU treated cells were more sensitive to IGF-1 or IGF-1R inhibitors than untreated ones. It is not clear whether this is directly related to changes in translational regulation, nevertheless, it suggests that targeting the IGF-1/IGF-1R pathway may improve 5-FU efficacy. This data also supports that the IGF-1/IGF-1R pathway might contribute to drug tolerance. In addition, several mRNAs coding for proteins carrying anti-apoptotic functions, metabolic functions and cellular differentiation functions were translationally upregulated in 5-FU treated cells, and may contribute to cellular treatment escape. Indeed, metabolic adaptation and phenotypic plasticity, have been associated with cellular drug tolerance, and the ability of a small number of cells to survive drug pressure in a quiescent or slow-proliferative state[4,48]. Our data further support that translation control could represent a non-genomic mechanism contributing to drug tolerance, and will require single-cell analysis[49,50]. 5-FU is the most widely used chemotherapy, and there is a high demand for improving its efficacy. Our data highlight the potential benefits of understanding drug-tolerance mechanisms in response to 5-FU, which has so far not been fully described. In addition, while our study focused on a base analogue incorporated into RNA, other compounds binding to RNAs such as platin derivatives or any drug that might interfere with RNA metabolism should now also be considered as modifiers of ribosome structure and activity[51], and may also contribute to altering translational regulation in treated cells.

Altogether, our study extends the spectrum of gene expression mechanisms that help cells survive a drug challenge, by adding translational regulation to epigenetics, stress response, metabolism adaptation and stemness or dormancy phenotypes[1,2,4,52–54]. These findings also reveal that exposure to drugs can result in the production of "man-made" biological complexes, the functioning of which cannot be anticipated, and that require further studies to fully comprehend drug response and propose new therapeutic strategies.

## Methods

**Cell lines, cell culture, and 5-FU treatment.** Cells were maintained in Dulbecco Minimum Essential Medium—GlutaMax (Invitrogen) supplemented with 10% foetal bovine serum (FBS) at 37 °C with 5% $CO_2$. The following cell lines were obtained from ATCC: HCT116 (ATCC CCL-247), MDA-MB-231 (ATCC HTB-26), BT20 (ATCC HTB-19), HT29 (ATCC HTB-38), SW480 (ATCC CCL-228) Panc1 (ATCC CRL-1469) and MiaPaCa (ATCC CRL-1420). The following cell lines were obtained from the authors: ISRECO1 (Cajot, J. F., et al. Cancer Res. (1997) 57, 2593–2597). The following cell lines were authenticated by 21 PCR-single-locus-technology (Eurofins, Ebersberg, Germany): HCT116, HT29, MDA-MB-231, MiaPaCa, Panc1. ISRECO1 cell line was not authenticated by PCR-single-locus-technology as this cell line genetic pattern is not described in databases. Cells were routinely tested against mycoplasma infection.

Cells were plated 48 h before 5-FU treatment. 5-FU was kindly provided by the Centre Léon Bérard (Lyon, FRANCE) and was purchased from Sanofi-Aventis. The stock solution was diluted immediately before use in DMEM.

**Western Blot.** Western blot was performed as previously reported[55]. Briefly, cells were counted and a defined number of cells were lysed in lysis buffer A (20 mM HEPES-KOH pH 7.2, 100 mM KCl, 1 mM DTT, 0.5 mM EDTA, 0.5% NP40, 10% Glycerol) supplemented with protease inhibitor (Complete EDTA free, Roche). A volume corresponding to 100,000 cells was loaded on a polyacrylamide gel, and

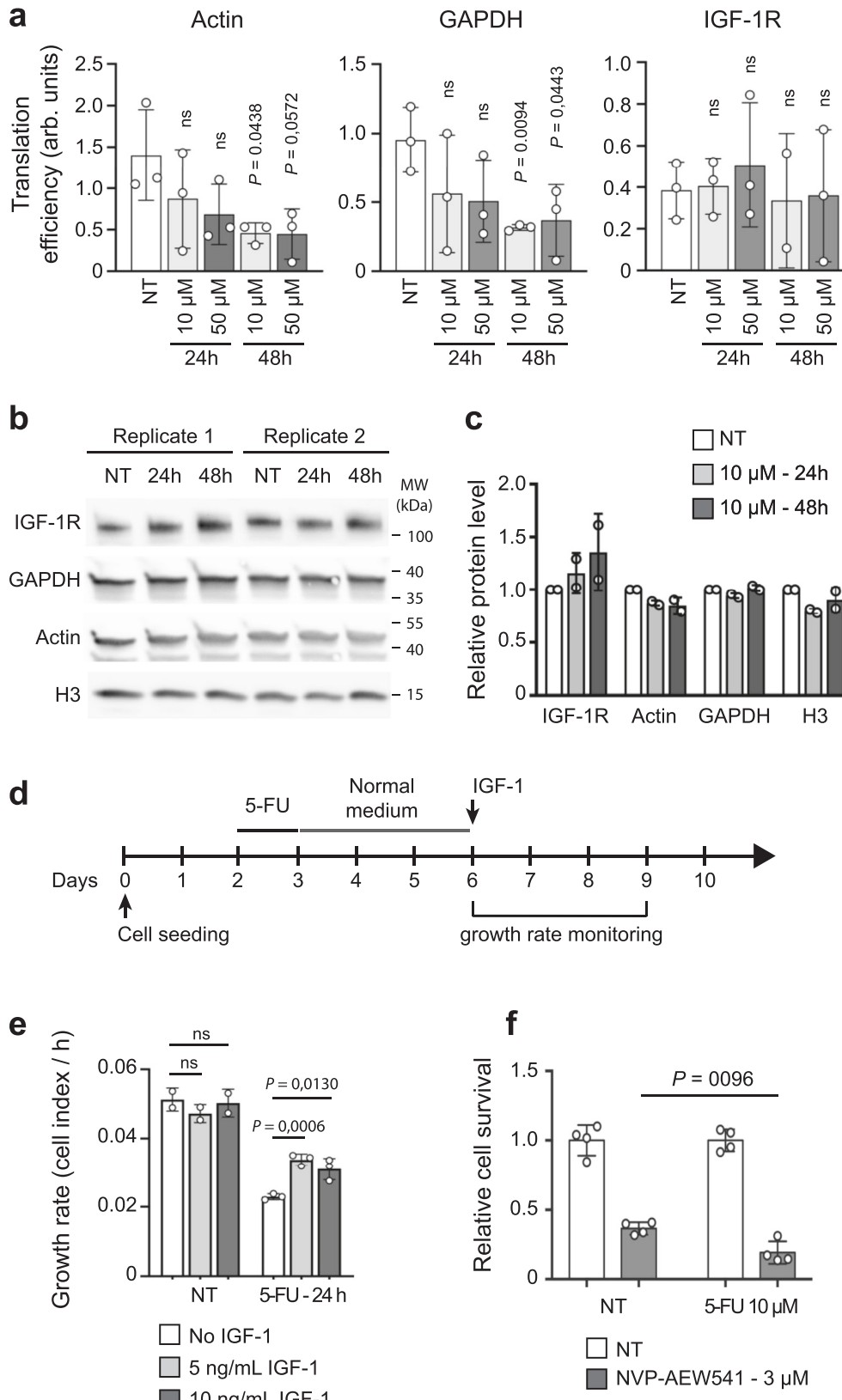

proteins were subjected to SDS-PAGE, and blotted onto poly-vinylidene difluoride (PVDF) membranes (Immobilon-P, Millipore). Membranes were blocked in Tris-buffered saline solution containing 0.05% Tween 20 and 5% nonfat milk and incubated with primary antibodies: mouse monoclonal antibodies against GAPDH [6C5] (AM4300, Invitrogen) and rabbit monoclonal antibodies against IGF-1R (#9750, Cell Signalling) and actin (ab179467, Abcam) and anti-Histone H3 (ab1791, Abcam)

diluted at 1:2000. Horseradish peroxidase-conjugated secondary antibodies Antirabbit IgG, HRP-linked Antibody (#7074, Cell Signalling Technologies) and Antimouse IgG, HRP-linked Antibody (#7076, Cell Signalling Technologies) diluted at 1:5000 were used for the detection of immunoreactive proteins by chemiluminescence (Clarity Western ECL Substrate, BioRad). Imaging and densitometric measurements of the bands were performed using Image lab software (BioRad).

**Fig. 4 IGF-1R contributes to survival and recovery of 5-FU treated CRC cells. a** HCT116 cells were treated with 10 μM or 50 μM 5-FU for 24 h or 48 h or untreated (NT). Translation efficiency of actin, GAPGH and IGF-1R mRNAs. Each mRNA was quantified from cytoplasmic and polysomal fractions. Translation efficiency are shown as the ratio of polysomic mRNA over the cytoplasmic mRNA. Each dot represents an individual biological sample measured in duplicate and data are expressed as mean ± s.d of independent experiments ($n = 3$). **b**, **c** HCT116 cells were treated with 10 μM 5-FU for 24 h or 48 h or untreated (NT). **b** Cells were counted and an equivalent number of cells were loaded in each well. IGF-1R, Actin, GAPDH and H3 proteins were detected by western blot. **c** Level of IGF-1R, Actin, GAPDH and H3 proteins quantified from the western blot in **b**. Signals of each protein was normalized to the untreated (NT) value. Each dot represents an individual biological sample and data are expressed as mean ± s.d of independent experiments ($n = 2$). **d**, **e** HCT116 cells were treated with 10 μM 5-FU for 24 h or 48 h or NT, and not stimulated (No IGF-1) or stimulated with 5 or 10 ng/mL of IGF-1. Cell growth was monitored in real-time over 5 days. **d** Schematic representation of the experiment. **e** Growth rate measured over 72 h (day 6 to day 9). Each dot represents a technical replicate and data are expressed as mean ± s.d. **f** HCT116 cells were untreated (NT) or treated with 10 μM of 5-FU alone or with 5 μM of IGF-1R inhibitor NVP-AEW541 alone or with 10 μM of 5-FU for 48 h. Cell survival was assessed using MTS at 72 h post-treatment. Each dot represents a technical replicate and data are expressed as mean ± s.d. of independent experiments ($n = 4$). Results of unpaired two-tailed $t$ test are indicated as nonsignificant (ns) $p < 0.05$ (*), $p < 0.01$ (**), $p < 0.001$ (***), and $p < 0.0001$ (****). Source data are provided as a Source Data file.

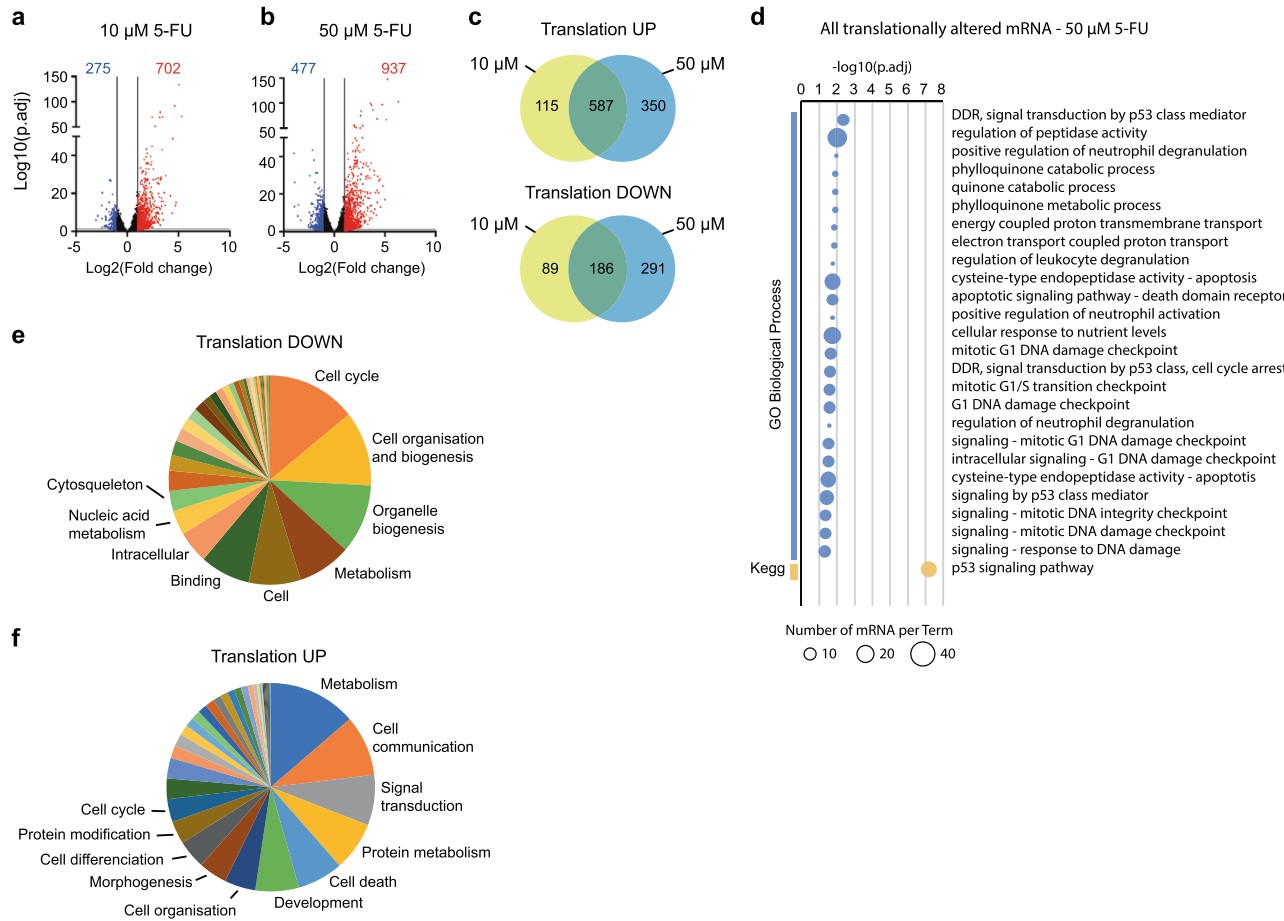

**Fig. 5 Alteration of specific mRNA translation by 5-FU.** HCT116 cells were untreated (NT) or treated with 10 μM or 50 μM 5-FU for 24 h and mRNA translation was assessed by RNA-sequencing of polysome-associated mRNAs. **a–b**, mRNA association with polysomes was compared between 10 μM (**a**) or 50 μM (**b**) 5-FU treated and untreated cells, and significance was tested using a two-sided Wald test from DESeq2 R package. Adjustments for multiple comparisons were made using Benjamini and Hochberg correction method. *P* values for individual mRNA are available in Supplementary Data 2. The data show all analyzed mRNAs. mRNAs whose polysomal level either increased (red) or decreased (blue) by more than twofold between untreated and 5-FU treated cells are colored (*p*.adj < 0.05). The threshold +/− 1 log2(FC) is indicated by vertical lines. The number of significantly altered mRNAs is indicated on top of the graphs. **c** Same datasets as in (**a**) and (**b**). Venn diagrams showing the comparison of translationally upregulated and downregulated mRNAs in 10 μM and 50 μM 5-FU treated cells. **d** Gene ontology analysis of all translationally altered mRNAs upon treatment with 50 μM 5-FU. The data show enriched GO terms for biological processes category (GO-Term-BP) and KEGG pathways. Analysis performed using gprofiler[63]. **e–f**, Significantly enriched GO terms were clustered in sub-groups according to GOslim2 classification and using with the CateGOrizer webservice[64]. The pie charts summarize the list of functional sub-groups in order of decreasing importance respectively for translationally downregulated genes (**e**) and translationally upregulated genes (**f**). The colours were chosen by order of importance, and do not represent the same groups in (**e**) and (**f**). Source data are provided as a Source Data file.

**Purification of cytoplasmic ribosomes.** Except otherwise stated, cytoplasmic ribosome purification from cells was performed as previously described[56]. Briefly, cytoplasmic fractions were obtained by mechanical lysis of cells using a Precellys tissue homogenizer (Ozyme) and centrifugation at 12,000 g for 10 min to pellet mitochondria. To purify ribosomes, cytoplasmic fractions were adjusted to 500 mM KCl by adding an appropriate volume of 3 M KCl, and loaded onto a 1 M sucrose cushion in a buffer containing 50 mM Tris-HCl pH 7.4, 5 mM MgCl$_2$, 500 mM KCl and 2 mM DTT, and centrifuged for 2 h at 240,000 g. The pellet containing the ribosomes was quickly washed with and resuspended in a buffer containing 50 mM Tris-HCl pH 7.4, 5 mM MgCl$_2$ and 25 mM KCl.

For puromycin treatment, cells were fractionated as described above. The cytoplasmic fraction was loaded onto a 25 mM KCl, 1 M sucrose cushion in a buffer containing 50 mM Tris-HCl pH 7.4, 5 mM MgCl$_2$ and 2 mM DTT, and centrifuged for 2 h at 240,000 g. The pellet containing the ribosomes was quickly washed and was resuspended in a buffer containing 50 mM Tris-HCl pH 7.4, 5 mM MgCl$_2$ and 25 mM KCl. Suspended ribosomes were treated with 1 µg/mL puromycin (Sigma) for 10 min at 4 °C and 15 min at 37 °C. Upon puromycin dissociation, the suspension was adjusted to 500 mM KCl with a 3 M KCl solution, and loaded onto a 1 M sucrose cushion in a buffer containing 50 mM Tris-HCl pH 7.4, 5 mM MgCl$_2$, 500 mM KCl and 2 mM DTT, and centrifuged for 2 h at 240,000 g. The pellet containing the ribosomes was quickly washed and was resuspended in a buffer containing 50 mM Tris-HCl pH 7.4, 5 mM MgCl$_2$ and 25 mM KCl.

**Ribosome purification from polysomal fraction.** Cells were seeded at 10[7] cells/ 15 cm dish and treated with 5-FU 48 h later as indicated. Cells were then incubated for 5 min with 25 µg/mL Emetin (Sigma) and washed twice with cold 1X PBS before harvesting. Cytosolic lysates were prepared as described above. 3 mg of cytosolic proteins was loaded onto a 10–40 % sucrose gradient, sedimented by ultracentrifugation for 2 h at 240,000 g at 4 °C on a SW-41 rotor (Beckman). Fractions were collected and absorbance profiles were generated at 245 nm using an ISCO UA-6 detector. Polysomal fractions were pooled and concentrated in an Amicon Ultra-15 unit with a 100 kDa cut-off. KCl concentration was adjusted to 500 mM using a 4 M stock solution. The ribosome suspension was reduced to 600 µL on an Amicon Ultra-15 filtration unit, and loaded onto a 1 M sucrose cushion in a buffer containing 50 mM Tris-HCl pH 7.4, 5 mM MgCl$_2$, 500 mM KCl and 2 mM DTT, and centrifuged for 2 h at 240,000 g. The pellet containing the ribosomes was resuspended in a buffer containing 50 mM Tris-HCl pH 7.4, 5 mM MgCl$_2$ and 25 mM KCl.

**rRNA purification.** For purified ribosomes, RNA was extracted using the TriPure Reagent (Roche) according to the manufacturer's instruction. Purified rRNAs were resuspended in water and quantified by spectrophotometry. For xenograft and human tumour samples, rRNAs were purified as described previously[24]. Briefly, 10 µg of total RNA was denatured in 50% formamide at 70 °C for 10 min, and separated on a 0.8% low-melting agarose gel in 0.5X TAE buffer. 18 S and 28 S rRNA were gel-purified using the NucleoSpin Gel and PCR Clean-up kit and NT1 buffer (Macherey Nagel) according to the manufacturer's instruction.

**5-FU analysis by LC-HRMS.** Purified rRNA (1 to 3 µg) was digested overnight at 37 °C with 270 units of Nuclease S1 (Promega) using the supplied buffer. Next, nucleotides were dephosphorylated by directly adding to the mix 5U of calf intestine phosphatase (New England Biolabs) in 100 mM Tris-HCl, 50 mM NaCl, 10 mM MgCl$_2$, 0.025% Triton® X-100. Digestion was carried out overnight at 37 °C, the digested mix was then stored at −80 °C. For LC-HRMS analysis, on the day of the analysis, 300 µL of a mixture of methanol/water (70/30; v/v) was added and the samples were vigorously vortexed following the addition of labelled internal standard. Samples were centrifuged for 5 min. at 13,000 g, the supernatants were separated and evaporated to dryness under nitrogen at 37 °C. The residues were then resuspended in water before injection into the mass spectrometer device. Analysis was carried out on a liquid chromatography coupled with a high-resolution mass spectrometer (Q-Exactive Plus Orbitrap -Thermo Scientific). Data acquisition was performed alternatively in negative and positive modes with the full scan mode (FS) at a resolution of 70,000 or 140,000 and with the parallel reaction monitoring mode (PRM) at a resolution of 17,500. Chromatographic separation of nucleosides was achieved on a Hypercarb® column ((2.1 mm×100 mm; 5 µm) (ThermoScientific)). The level of 5-FU per ribosome was calculated as the ratio of measured [5-FUrd] over the measured [A], [C] and [G], divided by the relative quantity of each nucleotide per ribosome.

**Translatome analysis by polysome profiling.** Cells were seeded at 10[6] cells/15 cm dish and treated with 5-FU 48 h later as indicated. Cells were then incubated for 5 min with 25 µg/mL Emetin (Sigma) and washed twice with cold 1X PBS before harvesting. Cytosolic lysates were prepared by incubation of cells 10 min in hypotonic buffer (10 mM Tris-HCl pH 7.4, 0.5 mM MgCl$_2$, 10 mM KCl, 1X Complete[TM] EDTA free protease inhibitor (Roche) and 10 u/mL RNAseOut[TM] (Invitrogen)) followed by addition of 0.5 % NP-40. Nuclei were pelleted by centrifugation 5 min at 750 g and mitochondria were pelleted by centrifugation at 12,000 g for 10 min. Four mg of cytosolic proteins was loaded onto a 10–50 %

sucrose gradient, sedimented by ultracentrifugation for 2 h at 240,000 g at 4 °C on a SW-41 rotor (Beckman). Fractions of 750 µL were collected and absorbance profiles were generated at 245 nm using an ISCO UA-6 detector.

RNAs were extracted from the fractions containing 2 to 9 ribosomes per RNA. 150 µL of each fraction was collected and pooled and RNA were extracted using TRIzol[TM] LS reagent (Invitrogen) following manufacturer instruction and suspended in RNAse-free water. For sequencing, RNA samples were processed using Stranded mRNA Prep kit (Illumina) according to manufacturer's protocol. Briefly, mRNA molecules containing polyA tails were capture by oligo(dT) magnetic beads. Then, purified mRNA was fragmented and copied into first-strand complimentary DNA (cDNA) using reverse transcriptase and random primers. In a second strand cDNA synthesis step, dUTP replaced dTTP to achieve strand specificity. In the final steps, adenine and thymine bases were added to fragment ends and adapters were ligated. The resulting library was purified and selectively amplified for sequencing. Next-generation sequencing was performed on NextSeq 500 System (Illumina) using high output 2x75bp Flowcell and NextSeq System Suite v2.2.0 (Illumina).

**Bioinformatic analysis of translatomic data.** Translatome libraries read quality was assessed using FastQC v0.11.9 (Babraham Institute, Cambridge, UK). Reads were filtered according to quality threshold Q35 and were trimmed of 4 and 2 bases at their start and end, respectively, using Cutadapt v3.2[57]. With Cutadapt we set the minimal length of trimmed reads at 50 nucleotides: all trimmed reads shorter than 50 were removed from the analysis. High-quality reads were then aligned using STAR v2.7.7[58], on the Homo sapiens reference genome, version GRCh38.dna.-primary_assembly release v100, and annotated with GRCh38.100 gtf Ensembl annotation file. Quantification of mapped reads was performed using HTSeq-count v0.13.5[59], using the following parameters: strand = reverse, mode = intersection-nonempty, typesearch = exon, id_attributes = gene_id, additional attributes = gene_name, and ignoring chimeric reads. Statistical differential analyses were performed between the control and each 5-FU conditions (10 or 50 µM) using the Wald test from DESeq2 R package[60]. For each dataset, the read counts were filtered with a minimum of 1 count per million per biological sample after size factors estimation (method RLE for Relative Log Expression normalisation), and then dispersion was estimated (using DESeq2). P value adjustment that corrects for multiple tests to lower the risk of false discovery was performed with the method of Benjamini and Hochberg[61]. Genes with corrected p values below 0.05 were kept. Gene identifications were performed with biomaRt R package[62]. Functional annotations were performed with gProfileR[63] using a g:SCS threshold < at 0.80, and Ensembl release v103. Go terms were then clustered using CateGOrizer[64] according to the Goslim2 classification in multiple mode, which is well suited for the general-purpose analyses.

Gene ontology classification of mRNA of Survival genes was performed using the "functional annotation clustering" tool of the Database for Annotation, Visualization and Integrated Discovery (DAVID) version 6.8, using default parameters[65].

**Immunofluorescence analysis.** Cells were grown on glass coverslips, fixed in 4% paraformaldehyde in phosphate-buffered saline (PBS) before permeabilization with 0.5% Triton X-100 in PBS. Fibrillarin, Dyskerin and Nucleolin were detected using the anti-FBL rabbit polyclonal antibody (ab5821, Abcam) diluted at 1:2,000, anti-DKC1 rabbit polyclonal (sc-48794, Santa Cruz Biotechnology) diluted at 1:500 and anti-NCL mouse monoclonal antibody [4E2] (ab13541, Abcam) at 1:4,000. Secondary Antibody, Alexa Fluor 488 (A-11001, Thermofisher Scientific), Goat anti-Rabbit IgG (H + L) Cross-Adsorbed Secondary Antibody, Alexa Fluor 488 (A-11008, Thermofisher Scientific), Goat anti-Mouse IgG (H + L) Cross-Adsorbed Secondary Antibody, Alexa Fluor 555 (A-21422, Thermofisher Scientific), and Goat anti-Rabbit IgG (H + L) Cross-Adsorbed Secondary Antibody, Alexa Fluor 555 (A-21428, Thermofisher Scientific) were used at 1:1000. Coverslips were mounted using the Fluoromount G mounting medium (EMS). Images were acquired on a Zeiss LSM 780 confocal microscope using a 63X Plan Apochromat immersion objective (NA 1.4), as a Z-stack (voxel size: 0.0634 × 0.0634 × 0.3155 µm). Final images were prepared by maximum intensity projection to display all nucleoli, using Zeiss ZEN Black software. As indicated, level images of actinomycin D treated cells was adjusted to correct for NCL dispersion. Images were cropped using ImageJ (https://imagej.net)[66].

**[32P] pulse-chase labelling.** Labelling of newly synthesised RNAs was performed as described previously[21]. Briefly, cells were grown in DMEM containing 10% FBS, and 5-FU containing medium for 24 h or 48 h or actinomycin D at 50 ng/mL for 3 h. Phosphate deprivation was performed by incubating cells for 1 h with phosphate-free DMEM (Invitrogen) containing 10% dialysed FBS. For labelling, cells were incubated with phosphate-free-DMEM containing 10% dialysed FBS and 14 µCi/mL [32P]-orthophosphate (Perkin Elmer) for 2 h. The medium was then replaced by isotope-free medium and cells were harvested directly in TriPure Reagent (Roche) 3 h post-labelling. Total RNA was extracted using TriPure Reagent standard procedure and dissolved in formamide. 1 µg of total RNA was denatured at 75 °C for 10 min and separated in a 1% agarose-formaldehyde-tricine/ triethanolamine gel. The gel was dried for 2 h at 80 °C under a vacuum. Labelled

rRNAs were visualised by autoradiography using ImageQuant TL software (GEHealthCare) on a Typhoon PhosphorImager, (GE HealthCare). Isotope signal was normalised to ethidium bromide signal for 28 S and 18 S rRNA bands, and quantified using ImageJ (build bad6864e55 - https://imagej.net).

**Preparation of mRNA-associated polysomes**. This was performed as described in[24]. Briefly, cells were seeded at $10^7$ cells/15 cm dish and treated with 5-FU 48 h later as indicated. Cells were then incubated for 5 min with 25 µg/mL emetin (Sigma) and washed twice with cold 1X PBS before harvesting. Cytosolic lysates were prepared as described above. Three mg of cytosolic proteins was loaded onto a 10–40 % sucrose gradient, sedimented by ultracentrifugation for 2 h at 240,000 g at 4 °C on a SW-41 rotor (Beckman). Fractions were collected and absorbance profiles were generated at 245 nm using an ISCO UA-6 detector. Fractions corresponding to the second half of the polysomes (heaviest polysomes) were pooled and RNAs were extracted with TriPure Reagent as described by the manufacturer (Roche).

**Quantitative RT-PCR**. Two hundred and fifty nanogram of total RNA were reverse transcribed using the M-MLV RT kit and random primers (Invitrogen), according to the manufacturer's instructions. Quantitative real-time PCR (qPCR) was carried out using the Light cycler 480 II real-time PCR thermocycler (Roche). Expression of mRNAs was quantified using LightCycler 480 SYBR Green I Master Mix (Roche). The primers were obtained by oligonucleotide synthesis (Eurogentec) and are described in Supplementary Table 2.

**Xenograft tumour model**. Mice were 7 weeks old female Hsd: Athymic Nude-Foxn1$^{nu}$ (Envigo). $1.5 \times 10^6$ HCT116 cells were subcutaneously injected into nude mice flank ($n = 5$). At day 14, tumours had reached an average of 160 mm$^3$. Mice received three injections (days 14, 18, and 22) with either vehicle ($n = 2$) or 5-FU (50 mg/kg, $n = 3$). 4 h after the last treatment, mice were sacrificed and tumours were collected for subsequent analysis. Total RNA was isolated using the TRI REAGENT protocol (SIGMA T9424) followed by a clean-up step (RNeasy Micro, QIAGEN). rRNA were gel-purified as described above. Animal experiments were performed within French guidelines for experimental animal studies, under DSV agreement A34-172-13. The maximal authorized tumour burden was 1500 mm$^3$, and was not exceeded during the course of the study.

**Colorectal human tumour samples**. Patient tumours were collected and snap frozen. Total RNA was isolated using the TRI REAGENT protocol (SIGMA T9424) followed by a clean-up step (RNeasy Micro, QIAGEN). rRNAs were gel-purified as described above. Human samples were used under clinical agreement #NCT01577511 under the authority of Nîmes Carrémeau University Hospital. All patients signed an informed consent.

**In vitro hybrid translation**. Hybrid in vitro translation assay was performed as described previously (35) and is summarized hereafter. After centrifugation of 1 mL of RRL for 2 h 15 min at 240,000 g, 900 µL of ribosome-free RRL (named S100) was collected, frozen and stored at −80 °C. The extent of ribosome depletion from reticulocyte lysate was checked by translating 27 nM of in vitro transcribed capped and polyadenylated globin-Renilla mRNA in the S100 RRL and validated when no luciferase activity could be detected. In parallel, transfected cells were lysed in hypotonic buffer R (HEPES 10 mM pH 7.5, CH$_3$CO$_2$K 10 mM, (CH$_3$CO$_2$)$_2$Mg 1 mM, DTT 1 mM) and potter homogenised (around 100 strokes). Cytoplasmic fraction was obtained by 13,000 g centrifugation for 10 min at 4 °C. The ribosomal pellet was then obtained by ultracentrifugation for 2 h 15 min at 240,000 g in a 1 M sucrose cushion and was rinsed three times in buffer R2 containing HEPES 20 mM, NaCl 10 mM, KCl 25 mM, MgCl$_2$ 1.1 mM, β-mercaptoethanol 7 mM and resuspended in 30 µL of buffer R2 to reach a ribosomal concentration exceeding 10 µg/µL for optimal and long-term storage at −80 °C. The reconstituted lysate was then assembled by mixing 5 µL of S100 RRL with a scale from 0.25 to 4 µg of ribosomal pellet. Typically, the standard reaction contained 5 µL of ribosome-free RRL with 1 µg ribosomal pellet in a final volume of 10 µL. Upon reconstitution, the translation mixture was supplemented with 75 mM KCl, 0.75 mM MgCl$_2$ and 20 µM amino-acid mix.

For in vitro translation assays, p0-Renilla vectors containing the β-globin, GAPDH 5′UTR, CrPV, DCV or EMCV IRESs were described previously[38]. mRNAs were obtained by in vitro transcription, using 1 µg of DNA templates linearized at the AflII sites, 20 U of T7 RNA polymerase (Promega), 40 U of RNAsin (Promega), 1.6 mM of each ribonucleotide triphosphate, 3 mM DTT in transcription buffer containing 40 mM Tris-HCl (pH 7.9), 6 mM MgCl$_2$, 2 mM spermidine and 10 mM NaCl. For capped mRNAs, the GTP concentration was reduced to 0.32 mM and 1.28 mM of m7GpppG cap analogue (for β-globin mRNA) or m7GpppA (for CrPV mRNA) (New England Biolabs) was added. The transcription reaction was carried out at 37 °C for 2 h, the mixture was treated with DNAse and the mRNAs were precipitated with ammonium acetate at a final concentration of 2.5 M. The mRNA pellet was then resuspended in 30 µL of RNAse-free water and mRNA concentration was determined by absorbance using the Nanodrop technology. mRNA integrity was checked by electrophoresis on nondenaturing agarose gel.

**In cell translation assays**. Bicistronic constructs were obtained by cloning of the full-length human IGF-1R 5′UTR followed by 21 nt of the IGF-1R coding sequence (NCBI NM_000875) between the LucR and Luc+ coding sequences in the pIRES vector (REF Marcel 2019). The IGF-1R sequence was produced by gene synthesis and cloned in frame with the Luc+ coding sequence (GenScript®, Netherland). All deletion of the IGF-1R 5′UTR (Supplementary Fig. 5b) were obtained starting from the full-length IGF-1R sequence (GenScript®) using the numbering from the NCBI NM_000875 sequence. Monocistronic constructs were obtained by sub-cloning of the full-length IGF-1R 5′UTR (IGF-1R Full) or the 1-953 (IGF-1R ΔIRES) or 954-1040 (IGF-1R IRES) in pFL-IV-Luc+ vector. pFL-IV-Luc+ vector was obtained by removing the LucR coding sequence from the pIRES vector.

Translation assays were performed as follow: cells were seeded at 1000 cells/well in 96-well plates. Plasmids were transfected 24 h later with X-tremeGENE HP™ reagent (Roche Diagnostics) at 1:10 ratio with 200 ng of DNA according to manufacturer instruction. 24 h after transfection, cells were treated with 5-FU at 10 µM or 50 µM for 24 h. Luciferase activities were measured using the Dual-Luciferase reagent (Promega) on Spark plate reader (Tecan). Results were obtained from background-subtracted values, and ratios were calculated for each well. Data were normalized to the untreated condition (NT).

**Cell growth and viability assays**. HCT116 were seeded onto 96-well plates at 3,000 cells/well. 48 h after seeding, cells were treated for 48 h with 10 µM 5-FU alone or in combination with 5 µM NVP-AEW541 (Sigma-Aldrich), or with DMSO as a control, and for an additional 72 h with 5 µM NVP-AEW541 alone, and cell viability was evaluated by MTT assay (Cell Titer Aqueous One Solution Cell Proliferation Assay, Promega) according to the manufacturer's protocol. Cell growth was monitored in real-time using the xCELLigence technology (ACEA Biosciences), based on electric impedance generated by cells attached to the well. Signals were normalised against the time obtained with IGF-1 (5 or 10 ng/mL (Peprotech)) treatment. The growth rate was calculated over 72 h as the slope under the curve.

**Statistics and reproducibility**. Except for the bioinformatic analysis of translatomic data detailed above, statistical analysis was performed using the Prism software (version 7.0. GraphPad). A two-tailed unpaired student t-test was used for evaluating significance. IF, FISH and TEM experiments were performed at least 3 times, and images are representative of several fields observed for each experiment.

**Reporting Summary**. Further information on research design is available in the Nature Research Reporting Summary linked to this article.

## Data availability

RNA sequencing data of translatome analysis generated in this study are available at the NCI Gene Expression Omnibus database under accession number GSE178839. Datasets of translatome analysis and polysome profiles are available as supplementary data. All data used to generate the figures and tables are provided as supplementary data and in the "Source data" file accompanying this paper. Source data are provided with this paper.

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

## Acknowledgements

We thank Sandra Ortiz (Cancer Research Center of Lyon) for helpful discussions. We thank E. Errazuriz (Centre d'Imagerie Quantitative Lyon-Est) and C. Vanbelle (plateau d'imagerie du CRCL) for technical help. We thank Véronique Pantesco and and Saïd

Assou from the transcriptomic facility of IRMB for RNA-sequencing. We thank Brigitte Manship for editing the manuscript. G.T. and S.N-S. were recipient of a Ph.D. fellowship from la Ligue Nationale Contre le Cancer and from "La Fondation pour la Recherche Médicale". M.G. was supported by ANR Grant "INVADE" PC201507. The project was supported by the "Ligue Contre le Cancer, comités départementaux de la Drôme, du Rhône, du Puy-de-Dôme, et de l'Allier", by Agence Nationale de la Recherche (grant RiboMeth - ANR-13-BSV8-0012, grant PLAsCAN - ANR-17-CONV-0002), by the Institut National du Cancer (INCa, grant FluoRib - PLBIO18-131) and by Programmes d'Actions Intégrées de Recherche (PAIR Sein, RiboTEM, ARC_INCa_LNCC_7625) to J-J.D. This work is supported by the ERiCAN program of Fondation MSD-Avenir (Reference DS-2018-0015). J.-J.D. is part of the DevWeCan Labex Laboratory. We thank the ATGC bioinformatic platform for hosting computational analyses (support from PIA France Génomique and from Institut Français de Bioinformatique (ANR-11-INBS-0013)). E.R. and J.R. were supported by the GEM Flagship project funded from Labex NUMEV (ANR-10-LABX-0020). E.R., N.D.V., A.V., J.P. and F.C. are CNRS research fellow. A.D., J.-J.D., and V.M. are Inserm research fellows.

## Author contributions

G.T., Z.B-I., B.P., J.G., C.M., A.V., S.N-S., A.G., M.-A.M., M.G., M.C-D., F.L., V.M., G.S., E.B., M.-A. A., J.B.V., C.B., J.P., and F.C. performed experiments. G.T., J.G., V.M., C.V., H.C.M., A.D., J.C.S., N.D.V., T.O., J.P., F.C., and J.-J.D. designed experiments. M.P. and M.B. provided human biopsies. J.R. and E.R., performed next-generation sequencing data primary and secondary analyses, and developed scripts for specific analyses. P.B., T.O., J.P., J.G., N.D.V., F.C., and J.-J.D. designed and supervised the study. F.C. and J.-J.D. wrote the paper.

## Competing interests

The authors declare no competing interests.
