## [Peer Review File · Nature Communications]

Alteration of ribosome function upon 5-fluorouracil treatment favors cancer cell drug-toleranceREVIEWER COMMENTS

Reviewer #1 (Remarks to the Author):

The manuscript by Therizols et al. suggests that 5-FU treatment induces the formation of fluorinated ribosomes that enhance the translation of survival genes. Although here presented data demonstrate an increase in fluorinated ribosomes after 5-FU treatment, the mechanism underlying the augmented translation of survival genes is missing. The authors should have integrated in their manuscript additional data, including a transcriptomic profile analysis, to determine the involvement of fluorinated ribosomes in survival gene translation.

The following are the concerns of this reviewer:

- In all the manuscript, with the exception of Figure 2, authors confirm their hypothesis by using only one established colorectal cancer cell line (HTC116). By contrast, in Figure 2D and E authors claim that fluorinated ribosomes are present in three tumor xenografts, generated by the established cell line HTC116, and in tumor cells from five 5-FU treated patients. In order to validate and strengthen the obtained results, authors should perform both in vitro and in vivo experiments using colorectal cancer cell lines with different mutational background and expand the collection of primary cancer cells .
- Based on the results obtained in Figure 3C, authors focus on mRNAs more efficiently translated by fluorinated ribosome and in particular on IGF-1R. For this referee it is not clear why the authors investigated the role of IGF-1R instead of c-MYC or other survival genes in promoting 5-FU drug tolerance. For instance, the treatment with 5-FU increased the translation of c-Myc more than IGF-1R. Authors should motivate their choice, since also c-Myc promotes the survival of colorectal cancer cells.
- To verify whether IGF-1 pathway sustains the survival of 5-FU treated HTC116 cells, authors evaluate the proliferation of untreated/treated cells in presence of IGF1 up to 72 hours. It is surprising that IGF-1, which has been demonstrated to promote colorectal cancer cell growth, has no effect on the proliferation rate of untreated cells.
- The authors claim that the inhibition of IGF-1R decreases the number of cells that survive to 5-FU treatment. However, they are not showing the effect of IGF-1R inhibition in absence of 5-FU, considering that it will also affect cell proliferation by itself.
- Statistical analysis is missing in several graphs.
- Authors should carefully check the references section in order to verify the exact match and to avoid repetitions. Indeed, reference number 19 and 38, and 20 and 39 are the same.

Reviewer #2 (Remarks to the Author):

In this manuscript Therizols and colleagues provide evidence that the chemotherapeutic agent 5-fluorouracil (5FU) incorporates into ribosomes and that 5FU-containing ribosomes selectively impact translation of certain mRNAs. While these observations are intriguing and relevant to cancer therapy, the data provided are not sufficiently compelling and do not fully support the authors' claims. In some instances, the data is in conflict with the authors' conclusions. In other cases, the lack of essential information (e.g. sucrose gradients traces, uncropped western blots and detail methods for 5FU-rRNA extraction and 5FU-ribosome purification) precluded the proper assessment of the data. Therefore, I conclude that in its current form the presented manuscript is not suitable for publication.

Specific comments:

1) It seems unlikely that a seemingly small number of randomly incorporated 5FU molecules can have such a drastic effect on translation, unless they are selectively incorporated on a critical region of the ribosome. This is not addressed in the manuscript.

2) rRNA purification strategy used in figure 2 and 3a does not remove mRNAs, and mRNAs with 5FU incorporated cannot be excluded from the LC-HRMS experiment. To show this more accurately, authors can perform EDTA treatment to disassemble translating ribosomes and release

mRNAs, then perform a sucrose cushion to isolate free subunits. 5-FU quantification by LC-HRMS can be performed on gel selected rRNAs from this prep.

3) It is unclear what leads to the discrepancy between enhanced translation of IGF-1R reporters in 3C by ribosomes from 5-FU treated cells and no change in translation of endogenous IGF-1R mRNA in 5-FU treated cells.

4) Unclear why translation efficiencies in 4a are reduced for actin and GAPDH, but not the protein levels of GAPDH in 4b. This is not acknowledged. No specific proteins are shown to change in expression upon 5-FU treatment. This makes it hard to interpret the lack of a change in IGF-1R translation, since there doesn't seem to be a relationship between translation efficiency and mature protein levels.

5) The proposed model is that 5-FU treatment causes global reduction in translation. However, certain mRNAs (e.g. IGF-1R) are resistant to this. Sustained IGF-1R expression promotes viability of 5-FU treated cells. This model suggests that 5-FU treated cells are more sensitive to IGF-1R inhibition. However, the authors do not inhibit IGF-1R with NVP-AEW541 in NT cells in figure 4E. Without this, we can't know if 5-FU cells are more sensitive to IGF-1R inhibition than NT cells, which should be true according to the model.

An Alternate model to explain results of figure 4d/e is that IGF-1 translation is inhibited when adding 5-FU, this would explain results in 4d/e. Adding more IGF-1 rescues this growth defect, and inhibiting IGF-1R exacerbates this effect.

Minor comments:

1) For the section titled "5-FU incorporation into ribosomes", on line 8/9 of that page, the text says to "see methods for rRNA purification strategy". However, it is ambiguous whether this is referring the reader to the methods section titled "Ribosome purification from whole cells", or "Ribosome purification from polysomal fraction".

2) Title of first section claims that 5-FU does not inhibit ribosome production. However mature levels of 18S and 28S rRNAs after 5-FU treatment are ~50% that of untreated cells. This is interpreted by the authors as being maintained at substantial levels, which seems like a subjective cutoff.

3) Figure 4B says normalized to actin, legend says normalized to Ku80.

4) Unclear what the arrows in Extended figure 3b are referring to.

Alteration of ribosome function upon 5-fluorouracil treatment favors cancer cell drug-tolerance

REVIEWER COMMENTS

We thank the reviewers for their critical comments and suggestions to improve our manuscript, in particular to consolidate the demonstration that 5-FU is incorporated in ribosomal RNA and that translation of genes contributing to cell tolerance is favored.

We made a major revision that led to additional experiments, including 1 new main figure and 1 new supplementary figure, 3 new panels, 5 new supplementary panels, and 4 modified main figure panels. We provide below a detailed point-by-point response and associated additional data for each comment raised by the reviewers.

Reviewer #1 (Remarks to the Author):

The manuscript by Therizols et al. suggests that 5-FU treatment induces the formation of fluorinated ribosomes that enhance the translation of survival genes. Although here presented data demonstrate an increase in fluorinated ribosomes after 5-FU treatment, the mechanism underlying the augmented translation of survival genes is missing. The authors should have integrated in their manuscript additional data, including a transcriptomic profile analysis, to determine the involvement of fluorinated ribosomes in survival gene translation.

Response: We thank the reviewer for her/his suggestions to improve our manuscript. We have now further strengthened our data demonstrating the incorporation of 5-FU in ribosomes. In addition, as suggested, we have added data on the mechanism underlying the translation regulation of IGF-1R mRNA by fluorinated ribosomes, by demonstrating an increased activity of IGF-1R internal ribosome entry site element (IRES) upon 5-FU treatment. This work includes a full mapping of IGF-1R 5'-UTR, and a demonstration that only the IRES domain displays an altered translation in 5-FU treated cells (See page 7, lines 3-18, Fig. 3e and 3f; Suppl. Fig. Fig. 4b, 4c, 4d).

As suggested, we performed a translome analysis by polysome profiling to get a genome wide view of survival gene mRNA translation and we show that upon 5-FU treatment, mRNAs of genes promoting cell survival, inhibition of apoptosis and cell phenotypic adaptability are more translated (Pages 9-10, Fig.5; Ext. Data fig. 6).

The following are the concerns of this reviewer:

- In all the manuscript, with the exception of Figure 2, authors confirm their hypothesis by using only one established colorectal cancer cell line (HTC116). By contrast, in Figure 2D and E authors claim that fluorinated ribosomes are present in three tumor xenografts, generated by the established cell line HTC116, and in tumor cells from five 5-FU treated patients. In order to validate and strengthen the obtained results, authors should perform both in vitro and in vivo experiments using colorectal cancer cell lines with different mutational background and expand the collection of primary cancer cells .

Response: We agree with the reviewer. We extended the experiments demonstrating incorporation of 5-FU in rRNA of two other cell lines, HT29 and SW480, that are characterized by different genetic backgrounds, in particular regarding KRAS, BRAF, p53 and the MSI status (we added Suppl. Table 1 for their description). The 3 cell lines show similar incorporation level of 5-FU in vitro and 5-FU was found in all tested xenograft of the hT29 and SW480 cell lines. The novel data are described in Fig.2c for cultured cell lines) and Fig. 2d-f for xenografts and the text was modified (Page 5, line 10-11).

We agree that it would be insightful to extend the number of primary tumor samples. However, CRC tumors are not routinely collected from patients treated with 5-FU and consequently are rarely available in tumor banks. Unfortunately, we were not able to have access to additional samples during the revision process. Nevertheless, the data we already collected does demonstrate that 5-FU can be found in rRNA of patient tumor cells.

- Based on the results obtained in Figure 3C, authors focus on mRNAs more efficiently translated by fluorinated ribosome and in particular on IGF-1R. For this referee it is not clear why the authors investigated the role of IGF-1R instead of c-MYC or other survival genes in promoting 5-FU drug tolerance. For instance, the treatment with 5-FU increased the translation of c-Myc more than IGF-1R. Authors should motivate their choice, since also c-Myc promotes the survival of colorectal cancer cells.

Response: The primary focus of this study is to demonstrate that 5-FU incorporates in ribosomes and alter translation in a way that may impact cellular phenotype. C-Myc and IGF-1R mRNAs were carefully selected both for their contribution to the cancer cell behavior and very importantly for their 5'UTR which are structured and support particular translation regulation. We focused on IGF-1R mRNA because its 5'UTR has been well described and contains several regulatory elements (structured domains, uORF, GC-rich regions and an IRES, see reference 29 in the MS) that could be molecularly tested, and has now been fully exploited as mentioned above. To clearly state our reasoning, the text was modified page 6, lines 7-10, and page 7, lines 3-5.

- To verify whether IGF-1 pathway sustains the survival of 5-FU treated HTC116 cells, authors evaluate the proliferation of untreated/treated cells in presence of IGF1 up to 72 hours. It is surprising that IGF-1, which has been demonstrated to promote colorectal cancer cell growth, has no effect on the proliferation rate of untreated cells.

Response: Indeed, we did notice that untreated cells did not proliferate more when treated with IGF-1. We have tested several concentrations of IGF-1 on HCT116 cultured in the presence of various serum concentration of serum or no serum, and we confirm that IGF-1 did not impact cell growth on our HCT116 cells.

- The authors claim that the inhibition of IGF-1R decreases the number of cells that survive to 5-FU treatment. However, they are not showing the effect of IGF-1R inhibition in absence of 5-FU, considering that it will also affect cell proliferation by itself.

Response: We thank the reviewer for commenting on this oversight. We have now included the missing control. The new figure (Fig. 4f) now contains all controls, and shows that IGF-1R inhibition has a stronger effect on cell survival upon 5-FU treatment as compared to untreated.

- Statistical analysis is missing in several graphs.

Response: We added statistical analysis to Fig. 4e, to Suppl. Fig. 3d (Ext. Data Fig. 3b in original manuscript) and to Suppl. Fig. 4a.

Statistical analyses have been included in all new panels of the revised manuscript.

- Authors should carefully check the references section in order to verify the exact match and to avoid repetitions. Indeed, reference number 19 and 38, and 20 and 39 are the same.

Response: references have been corrected.

Reviewer #2 (Remarks to the Author):

In this manuscript Therizols and colleagues provide evidence that the chemotherapeutic agent 5-fluorouracil (5FU) incorporates into ribosomes and that 5FU-containing ribosomes selectively impact translation of certain mRNAs. While these observations are intriguing and relevant to cancer therapy, the data provided are not sufficiently compelling and do not fully support the authors' claims. In some instances, the data is in conflict with the authors' conclusions. In other cases, the lack of essential information (e.g. sucrose gradient traces, uncropped western blots and detail methods for 5FU-rRNA extraction and 5FU-ribosome purification) precluded the proper assessment of the data. Therefore, I conclude that in its current form the presented manuscript is not suitable for publication.

We thank the reviewer for her/his suggestions to improve our manuscript. We have included all source data including sucrose gradient profiles (Supp. Data 1, 2 and 3) and uncropped images of western blot (see source data). The method section describing purification of cytoplasmic ribosomes have been completed and now describes puromycin-based subunit dissociation (see point #2 below). All methods of ribosome purification and rRNA extraction are described in details and methods are now linked to each type of sample used in the study (whole cells, xenograft, human samples, polysomal fractions).

Specific comments:

1) It seems unlikely that a seemingly small number of randomly incorporated 5FU molecules can have such a drastic effect on translation, unless they are selectively incorporated on a critical region of the ribosome. This is not addressed in the manuscript.

Response: The point raised by the reviewer is a key question of ribosome biology that concerns not only our study with 5-FU but also any chemical modification of rRNA. Our reasoning is indeed based on a preferential enrichment of 5-FU in regions of rRNA where 5-FU would not have deleterious effect on rRNA processing and ribosome assembly. We modified the text of the discussion to state this point (Page 10, lines 21-22 and 25-26) and also to illustrate which regions of rRNA might be critical for IRES-dependent translation that is now described in the manuscript (Page 11, lines 13-18). Unfortunately, for the time being, the analytical tools available do not allow for a precise localization of 5-FU within each in rRNA molecule.

2) rRNA purification strategy used in figure 2 and 3a does not remove mRNAs, and mRNAs with 5FU incorporated cannot be excluded from the LC-HRMS experiment. To show this more accurately, authors can perform EDTA treatment to disassemble translating ribosomes and release mRNAs, then perform a sucrose cushion to isolate free subunits. 5-FU quantification by LC-HRMS can be performed on gel selected rRNAs from this prep.

Response: The reviewer is right, rRNA purification performed directly on 500 mM KCl sucrose cushion might result in minor mRNA contamination. To exclude that 5-FU detection was due to contaminating RNAs, we first used gel purified rRNA as shown in Suppl. Fig. 3a of the original manuscript. As suggested by the reviewer, we had performed puromycin treatment to fully dissociate subunits. The new Suppl. Fig. Fig 3b and 3c shows the profiles of ribosomal subunit that we obtained with both methods, and which reveals a very small contamination of 80S ribosomes using 500 mM KCl sucrose cushion only, and that disappear upon puromycin subunit dissociation. We then analyzed 5-FU incorporation in rRNA of ribosomes purified with both methods, on HCT116 cells lines and on the two

additional cell lines HT29 and SW480. The new *Suppl. Fig 3c* shows that 5-FU incorporation is identical with both methods, on all three cell lines, treated with either 10 μ M or 50 μ M 5-FU.

Thus 5-FU incorporation appears solely due to rRNA incorporation.

The main text (*page 4, lines 15-23*) and the method section (*page 3, lines 12-21*) have been modified accordingly.

3) It is unclear what leads to the discrepancy between enhanced translation of IGF-1R reporters in 3C by ribosomes from 5-FU treated cells and no change in translation of endogenous IGF-1R mRNA in 5-FU treated cells.

Response: In our view there is no discrepancy. The difference between Fig 3c and 4a is likely due to the difference in approach. Indeed, the method used in Fig. 3c aims at highlighting the change in behavior of ribosomes of different origins. In these experiments only ribosomes came from 5-FU treated cells, and none of the other components (eIF, mRNA...) have been in contact with 5-FU. In contrast, Fig. 4a reveals the behavior of endogenous mRNAs (not a reporter mRNA) in 5-FU treated cells, meaning that several cellular components (mRNA, tRNA) and not only ribosomes, might have been altered by 5-FU.

Thus, the data from both figures are not incompatible or contradictory. In addition, it remains that IGF-1R mRNA displays a higher translation activity than Actin or GAPDH mRNAs.

4) Unclear why translation efficiencies in 4a are reduced for actin and GAPDH, but not the protein levels of GAPDH in 4b. This is not acknowledged. No specific proteins are shown to change in expression upon 5-FU treatment. This makes it hard to interpret the lack of a change in IGF-1R translation, since there doesn't seem to be a relationship between translation efficiency and mature protein levels.

*Response: we agree that the lack of a clear difference in protein level seemed inconsistent with Fig. 4a. We had initially observed a tendency for IGF-1R to increase in 5-FU treated cells (that can be seen in the original Fig. 4b). In the process of the revision, we realized that any substantial difference in protein level would be compensated by loading a constant protein quantity in western blot. We thus repeated the experiment and loaded a constant number of cells per well (the method section was updated, see *page 2 of supplementary information*). This allows to determine changes in individual protein levels per cell unit. In the new figure *Fig. 4b and 4c*, we now show both the western blot images and associated quantification showing increase of IGF-1R levels and either stable or slight decrease of Actin and GAPDH levels. Actin and GAPDH proteins having long half-life, it is not expected to observe a significant decrease in protein levels upon short time period.*

5) The proposed model is that 5-FU treatment causes global reduction in translation. However, certain mRNAs (e.g. IGF-1R) are resistant to this. Sustained IGF-1R expression promotes viability of 5-FU treated cells. This model suggests that 5-FU treated cells are more sensitive to IGF-1R inhibition. However, the authors do not inhibit IGF-1R with NVP-AEW541 in NT cells in figure 4E. Without this, we can't know if 5-FU cells are more sensitive to IGF-1R inhibition than NT cells, which should be true according to the model.

An Alternate model to explain results of figure 4d/e is that IGF-1 translation is inhibited when adding 5-FU, this would explain results in 4d/e. Adding more IGF-1 rescues this growth defect, and inhibiting IGF-1R exacerbates this effect.

Response: While the reasoning of the reviewer is sound, our data does support our model that IGF-

1R/IGF-1 pathway plays a more important role in cell survival upon 5-FU treatment compared to untreated cells. Indeed, the **figure 4e** (4d in the initial submission) shows that 5-FU treated cells become sensitive to IGF-1 while the untreated are not. As addressed in reviewer #1, point #3 regarding untreated HCT116 cells sensitivity to IGF-1 stimulation, we have tested several IGF-1 concentrations and several serum concentrations in the culture medium and confirmed that untreated HCT116 cells are insensitive to IGF-1. Yet the cells become sensitive to IGF-1 once treated with 5-FU.

As for Figure 4e (**Fig. 4f** in the revised version), HCT116 treated only with the IGF-1R inhibitor was lacking in the initial version of the manuscript. It has now been added (see **Fig. 4f**) and shows that cells treated with 5-FU are more sensitive to IGF-1R inhibitor than cells not treated with 5-FU (81% inhibition vs 63% inhibition).

Altogether, these data show that cells become more sensitive or dependent to IGF-1/IGF-1R pathway once treated with 5-FU, and thus supports our hypothesis.

Minor comments:

1) For the section titled "5-FU incorporation into ribosomes", on line 8/9 of that page, the text says to "see methods for rRNA purification strategy". However, it is ambiguous whether this is referring the reader to the methods section titled "Ribosome purification from whole cells", or "Ribosome purification from polysomal fraction".

Response: this has been corrected and the method paragraph is now named "purification of cytoplasmic ribosomes" (Page 3, line 4 of the supplementary information). The method section describing ribosome purification and rRNA purification have been completed and linked to each type of sample used in the study.

2) Title of first section claims that 5-FU does not inhibit ribosome production. However mature levels of 18S and 28S rRNAs after 5-FU treatment are ~50% that of untreated cells. This is interpreted by the authors as being maintained at substantial levels, which seems like a subjective cutoff.

Response: We agree that the title of this section was misleading and we modified it as follow: "Ribosome production is partially maintained upon 5-FU treatment". See page 3, line 7.

3) Figure 4B says normalized to actin, legend says normalized to Ku80.

Response: The figure 4b has been modified and values are now normalized on NT signal.

4)Unclear what the arrows in Extended figure 3b are referring to.

Response: Arrows indicate 5-FU treatments of the animals. The information was missing and has been added in the legend.

REVIEWER COMMENTS

Reviewer #1 (Remarks to the Author):

Therizols et al. study provides evidence on the production of fluorinated ribosomes, which boost the translation of survival-associated genes inducing a drug-tolerance phenotype upon 5-FU treatment in colorectal cancer (CRC) cells. Although the manuscript has been largely improved and the authors addressed the majority of the reviewer concerns, the number of primary CRC samples is still limited in order to add translational clinical significance to the findings obtained by the analysis of established CRC cell lines.

Additionally, the authors should address the following concerns:

-The authors should generalize the data obtained by using the established CRC cell lines. Is there any difference in the transcriptome analysis of CRC patients resistant or sensitive to 5-FU treatment? Is there any correlation between the mutational background of the 5-FU resistant or sensitive CRC patients and the transcriptional levels of the survival genes identified by the authors?

- In Fig. 2d-f in both the Results section and in the Figure legend the authors should provide information about the samples M1, M2, M3.

Reviewer #2 (Remarks to the Author):

The authors have largely addressed my previous concerns.

Alteration of ribosome function upon 5-fluorouracil treatment favors cancer cell drug-tolerance

REVIEWER COMMENTS

We thank the reviewers for their additional comments and suggestions to improve our manuscript. We made additional significant modifications regarding the clinical significance of our data. We hope that the reviewers will now find our manuscript suitable for publication

Reviewer #1 (Remarks to the Author):

Therizols et al. study provides evidence on the production of fluorinated ribosomes, which boost the translation of survival-associated genes inducing a drug-tolerance phenotype upon 5-FU treatment in colorectal cancer (CRC) cells. Although the manuscript has been largely improved and the authors addressed the majority of the reviewer concerns, the number of primary CRC samples is still limited in order to add translational clinical significance to the findings obtained by the analysis of established CRC cell lines.

Additionally, the authors should address the following concerns:

-The authors should generalize the data obtained by using the established CRC cell lines. Is there any difference in the transcriptome analysis of CRC patients resistant or sensitive to 5-FU treatment? Is there any correlation between the mutational background of the 5-FU resistant or sensitive CRC patients and the transcriptional levels of the survival genes identified by the authors?

Response: We thank the reviewer for acknowledging and understanding the great difficulty to provide additional clinically relevant data at this stage, considering the scarcity of CRC samples collected post-treatment, and the lack of relevant data in public databases and in published datasets.

We fully understand the reviewer's comment regarding the small number of CRC samples that were analyzed. Accordingly, we acknowledge that the clinical and translational relevance of our data should be better balanced in the manuscript.

Consequently, we have modified the manuscript to soften the clinical and translational relevance of our data. In addition, the discussion section (page 11) now contains a paragraph discussing the interpretation of our data on the tumor samples.

- In Fig. 2d-f in both the Results section and in the Figure legend the authors should provide information about the samples M1, M2, M3.

Response: We modified the main text (page 5, line 11), and the Figure legend (Page 21, line 21).

Reviewer #2 (Remarks to the Author):

The authors have largely addressed my previous concerns.

Response: We are glad that we were able to address the reviewers' concerns.

REVIEWERS' COMMENTS

Reviewer #1 (Remarks to the Author):

The authors addressed the reviewer concerns. A paragraph pointing out the limitation of the data regarding patient samples was included in the discussion section.

NCOMMS-20-21031B

Alteration of ribosome function upon 5-fluorouracil treatment favors cancer cell drug-tolerance

Reviewer #1 (Remarks to the Author):

The authors addressed the reviewer concerns. A paragraph pointing out the limitation of the data regarding patient samples was included in the discussion section.

Authors response: We thank the reviewer for its positive comment.